# Automatic Data Augmentation via Invariance-Constrained Learning

## Abstract

Underlying data structures, such as symmetries or invariances to transformations, are often exploited to improve the solution of learning tasks. However, embedding these properties in models or learning algorithms can be challenging and computationally intensive. Data augmentation, on the other hand, induces these symmetries during training by applying multiple transformations to the input data. Despite its ubiquity, its effectiveness depends on the choices of which transformations to apply, when to do so, and how often. In fact, there is both empirical and theoretical evidence that the indiscriminate use of data augmentation can introduce biases that outweigh its benefits. This work tackles these issues by automatically adapting the data augmentation while solving the learning task. To do so, it formulates data augmentation as an invariance-constrained learning problem and leverages Monte Carlo Markov Chain (MCMC) sampling to solve it. The result is a practical algorithm that not only does away with a priori searches for augmentation distributions, but also dynamically controls if and when data augmentation is applied. Our experiments illustrate the performance of this method, which achieves state-of-the-art results in automatic data augmentation benchmarks for CIFAR datasets. Furthermore, this approach can be used to gather insights on the actual symmetries underlying a learning task.

## 1 Introduction

Exploiting the underlying structure of data has always been a key principle in data analysis. Its use has been fundamental to the success of machine learning solutions, from the translational equivariance of convolutional neural networks (Fukushima and Miyake, 1982) to the invariant attention mechanism in Alphafold (Jumper et al., 2021). However, embedding invariances and symmetries in model architectures is hard in general and when possible, often incurs a high computational cost. This is the case, of rotation invariant neural network architectures that rely on group convolutions, which are feasible only for small, discrete transformation spaces or require coarse undersampling due to their high computational complexity (Cohen and Welling, 2016; Finzi et al., 2020).

A widely used alternative consists of modifying the data rather than the model. That is, to *augment* the dataset by applying transformations to samples in order to induce the desired symmetries or invariances during training. Data augmentation, as it is commonly known, is used to train virtually all state-of-the-art models in a variety of domains (Shorten and Khoshgoftaar, 2019). This empirical success is supported by theoretical results showing that, when the underlying data distribution is invariant to the applied transformations, data augmentation provides a better estimation of the statistical risk (Chen et al., 2019; Sannai et al., 2019; Lyle et al., 2020; Shao et al., 2022). On the other hand, applying the wrong transformations can introduce biases that may outweigh these benefits (Chen et al., 2019; Shao et al., 2022).

Choosing which transformations to apply, when to do so, and how often, is thus paramount to achieving good results. However, it requires knowledge about the underlying distribution of the data that is typically unavailable. Several approaches to learning an augmentation policy or distribution over a fixed set of transformations exist, such as reinforcement learning (Cubuk et al., 2018), genetic algorithms (Ho et al., 2019), density matching (Lim et al., 2019; Cubuk et al., 2020; Hataya et al., 2020), gradient matching (Zheng et al., 2022), bi-level optimization (Li et al., 2020b; Liu et al., 2021), jointly optimizing over transformations using regularised objectives (Benton et al., 2020),

variational bayesian inference (Chatzipantazis et al., 2021) , bayesian model selection (Immer et al., 2022) and alignment regularization (Wang et al., 2022). Optimization based methods often require computing gradients with respect to transformations (Chatzipantazis et al., 2021; Li et al., 2020b). Moreover, several methods resort to computationally intensive search phases, the optimization of auxiliary models, or additional data, while failing to outperform fixed user-defined augmentation distributions (Müller and Hutter, 2021).

In this work, we formulate data augmentation as an invariance-constrained learning problem. That is, we specify a set of transformations and a desired level of invariance, and recover an augmentation distribution that enables imposing this requirement on the learned model, without explicitly parametrising the distribution over transformations. In addition, the constrained learning formulation mitigates the potential biases introduced by data augmentation without doing away with its potential benefits. More specifically, we rely on an *approximate* notion of invariance that is weighted by the probability of each data point. Hence, we require the output of our model to be stable only on the support of the underlying data distribution, and more so on common samples. By imposing this requirement as a constraint on the learning task and leveraging recent duality results, the amount of data augmentation can be automatically adjusted during training. We propose an algorithm that combines stochastic primal-dual methods and MCMC sampling to do away with the need for transformations to be differentiable. Our experiments show that it leads to state-of-the-art results in automatic data augmentation benchmarks in CIFAR datasets.

## 2   DATA AUGMENTATION IN SUPERVISED LEARNING

As in the standard supervised learning setting, let $\mathbf{x} \in \mathcal{X} \subseteq \mathbb{R}^d$ denote a feature vector and $y \in \mathcal{Y} \subseteq \mathbb{R}$ its associated label or measurement. For classification tasks, we take $\mathcal{Y} \subseteq \mathbb{N}$. Let $\mathfrak{D}$ denote a probability distribution over the data pairs $(\mathbf{x}, y)$ and $\ell : \mathcal{Y} \times \mathcal{Y} \to \mathbb{R}_+$ be a non-negative, convex loss function, e.g., the cross entropy loss. Our goal is to learn a predictor $f_{\boldsymbol{\theta}} : \mathcal{X} \to \mathcal{Y}$ in some hypothesis class $\mathcal{H}_{\boldsymbol{\theta}} = \{f_{\boldsymbol{\theta}} \mid \boldsymbol{\theta} \in \Theta \subseteq \mathbb{R}^p\}$ that minimizes the expected loss, namely

$$\underset{\boldsymbol{\theta} \in \Theta}{\text{minimize}} \ R(f_{\boldsymbol{\theta}}) := \mathbb{E}_{(\mathbf{x}, y) \sim \mathfrak{D}}[\ell(f_{\boldsymbol{\theta}}(\mathbf{x}), y)]. \qquad \text{(SRM)}$$

We consider the distribution $\mathfrak{D}$ to be unknown, except for the dataset $\{(\mathbf{x}_i, y_i), i = 1, \dots, n\}$ of $n$ i.i.d. samples from $\mathfrak{D}$. Therefore, we rely on the empirical approximation of the objective of (SRM), explicitly

$$\hat{R}(f_{\boldsymbol{\theta}}) := \frac{1}{N} \sum_{i=1}^{n} \ell(f_{\boldsymbol{\theta}}(\mathbf{x}_i), y_i). \qquad (1)$$

One of the aims of data augmentation is to improve the approximation $\hat{R}$ of the statistical risk $R$ when dealing with a dataset that is not sufficiently representative of the data distribution. To do so, we consider transformations of the feature vector $g : \mathcal{X} \to \mathcal{X}$, taken from the (possibly infinite) transformation set $\mathcal{G}$. Common examples include rotations and translations in images. Data augmentation leverages these transformations to generate new data pairs $(g\mathbf{x}, y)$ by sampling transformations according to a probability distribution $\mathfrak{G}$ over $\mathcal{G}$, leading to the learning problem

$$\underset{\boldsymbol{\theta} \in \Theta}{\text{minimize}} \ \hat{R}_{\text{aug}}(f_{\boldsymbol{\theta}}) := \frac{1}{N} \sum_{i=1}^{N} \mathbb{E}_{g \sim \mathfrak{G}} \left[ \ell(f_{\boldsymbol{\theta}}(g\mathbf{x}_i), y_i) \right]. \qquad (2)$$

Note that the empirical risk approximation $\hat{R}$ in (1) can be interpreted as an approximation of the data distribution $\mathfrak{D}$ by a discrete distribution that places atoms on each data point. In that sense, $\hat{R}_{\text{aug}}$ in (2) can be thought of as the Vicinal Risk Minimization (Chapelle et al., 2000) counterpart of (1), in which the atoms on $\mathbf{x}_i$ are replaced by a local distribution over the transformed samples $g\mathbf{x}_i$, i.e.,

$$\hat{R}_{\text{aug}}(f_{\boldsymbol{\theta}}) = \frac{1}{N} \sum_{i=1}^{N} \int \ell(f_{\boldsymbol{\theta}}(g\mathbf{x}_i), y_i) \, dP(g\mathbf{x}_i), \qquad (3)$$

where the distribution $P$ over $\mathcal{X}$ is induced by the distribution $\mathfrak{G}$ over $\mathcal{G}$. As it can be seen from (3) if $\mathfrak{G}$ is not chosen adequately, $\hat{R}_{aug}$ can be a poor estimate of $R$, introducing biases that outweigh

the benefits of data augmentation (Chen et al., 2019; Shao et al., 2022). On the other hand, if the data distribution $\mathfrak{D}$ is statistically invariant under the action of $\mathcal{G}$, invariant solutions have provable advantages in terms of sample complexity (Chen et al., 2019; Bietti and Mairal, 2019; Sannai et al., 2019; Lyle et al., 2020).

In this work, given a set of transformations $\mathcal{G}$, we tackle the choice of $\mathfrak{G}$, i.e., how to sample transformations so that the solution of the learning problem is sufficiently invariant with respect to the learning task, as defined on the next section. Note that invariance to transformations in $\mathcal{G}$ may hold for the true distribution or may also be a desirable property of the solution, for example, to achieve robustness (Kanbak et al., 2017; Volpi et al., 2018; Joshi et al., 2019). Unlike *invariance learning* (Jebara, 2003; Zhou et al., 2021a; Benton et al., 2020; Immer et al., 2022), we do not seek to learn the transformations $\mathcal{G}$ from the data.

## 3 DATA AUGMENTATION VIA INVARIANCE CONSTRAINTS

Finding a transformation distribution $\mathfrak{G}$ that leads to the desired properties in the solution can be challenging . What is more, using a fixed $\mathfrak{G}$ as in (2) prevents us from controlling when and how much augmentation is used during training, running the risk of biasing the final solution. On the other hand, it is straightforward to specify a set of transformations $\mathcal{G}$ to which the solution should be *approximately* invariant (e.g., image rotations and translations).

In the next sections, we explain how a data augmentation distribution can be obtained from such an invariance requirement. We first show how *invariance* can be interpreted as an augmentation distribution (Section 3.1). We then incorporate this invariance in a constrained learning problem (Section 3.2).

### 3.1 FROM INVARIANCE TO AN AUGMENTATION DISTRIBUTION

The goal of this section is to replace the task of choosing an augmentation distribution by the task of choosing a set of transformations we wish to be (approximately) invariant to. To do so, we we will show how invariance can be used to implicitly determine the augmentation distribution.

However, rather than requiring the output of the model to be invariant, i.e., $f_{\boldsymbol{\theta}}(\mathbf{x}) = f_{\boldsymbol{\theta}}(g\mathbf{x})$ for all $g \in \mathcal{G}$, we will consider invariance in terms of quality of its predictions as evaluated by the loss function. Namely,

$$\ell(f_{\boldsymbol{\theta}}(\mathbf{x}), y) = \ell(f_{\boldsymbol{\theta}}(g\mathbf{x}), y), \ \forall g \in \mathcal{G}.$$

This notion of invariance explicitly incorporates the structure of the learning task by using the loss to identify which changes in the output of the model would lead to a significant change in prediction performance. More precisely, we want to limit the difference in our model's performance on a sample and its transformed versions, i.e., we wish to have $|\ell(f_{\boldsymbol{\theta}}(\mathbf{x}), y)) - \ell(f_{\boldsymbol{\theta}}(g\mathbf{x}), y))|$ small for all $g$ in $\mathcal{G}$. Equivalently, we wish to control the magnitude of

$$\max_{g \in \mathcal{G}} |\ell(f_{\boldsymbol{\theta}}(\mathbf{x}), y) - \ell(f_{\boldsymbol{\theta}}(g\mathbf{x}), y)| . \tag{4}$$

However, rather controlling (4) for all $x \in \mathcal{X}$, which may be overly conservative, we want to restrict our attention to the support of the data distribution. Furthermore, we wish to weight different inputs depending on their probability, in order to reduce the importance of unlikely or pathological cases. We therefore average (4) over the data distribution to obtain

$$R_{\mathrm{inv}}(f_{\boldsymbol{\theta}}) := \mathbb{E}_{(\mathbf{x},y) \sim \mathfrak{D}} \left[ \max_{g \in \mathcal{G}} |\ell(f_{\boldsymbol{\theta}}(\mathbf{x}), y) - \ell(f_{\boldsymbol{\theta}}(g\mathbf{x}), y)| \right] . \tag{5}$$

To connect the invariant risk $R_{\mathrm{inv}}$ in (5) to the data augmentation formulation in (2), notice that it can be bounded, using triangle inequality and the monotonicity of the expectation, by

$$R_{\mathrm{inv}}(f_{\boldsymbol{\theta}}) \leq \mathbb{E}_{(\mathbf{x},y) \sim \mathfrak{D}} [\ell(f_{\boldsymbol{\theta}}(\mathbf{x}), y)] + \mathbb{E}_{(\mathbf{x},y) \sim \mathfrak{D}} \left[ \max_{g \in \mathcal{G}} \ell(f_{\boldsymbol{\theta}}(g\mathbf{x}), y) \right] . \tag{6}$$

The first term is simply the statistical risk $R$ that is usually minimized in order to tackle the learning task. The second term resembles the objective typically found in adversarial learning problems, e.g., (Madry et al., 2017). This term, it turns out, can be interpreted as an augmentation distribution.

As shown by Robey et al. (2021a), the maximisation of the loss over transformations can be written as the semi-infinite constrained optimization problem

$$\max_{g \in \mathcal{G}} \ell \left( f_{\boldsymbol{\theta}} \left( g \mathbf{x} \right), y \right) = \sup_{\lambda \in \mathcal{L}_+^2} \int_{\mathcal{G}} \lambda(g) \ell(f_{\boldsymbol{\theta}}(g\mathbf{x}), y) dg. \tag{7}$$

$$\text{s. to } \int_{\mathcal{G}} \lambda(g) dg = 1$$

Notice that the solution of this optimization problem $\lambda^\star(g)$ is a non-negative, normalized function and can therefore be interpreted as a distribution over transformations that depends on the sample point $(\mathbf{x}, y)$ as well as the model $f_{\boldsymbol{\theta}}$. This allows us to re-interpret the maximization over $\mathcal{G}$ as an expectation, i.e.,

$$\mathbb{E}_{(\mathbf{x},y)\sim\mathfrak{D}} \left[ \max_{g \in \mathcal{G}} \ell(f_{\boldsymbol{\theta}} \left( g\mathbf{x} \right), y) \right] = \mathbb{E}_{(\mathbf{x},y)\sim\mathfrak{D}} \left[ \mathbb{E}_{g \sim \lambda^\star} \left[ \ell(f_{\boldsymbol{\theta}}(g\mathbf{x}), y) \right] \right]. \tag{8}$$

Observe that the right-hand side of (8) resembles the statistical form of the data augmentation objective in (2).

## 3.2 A Constrained Learning Approach

Returning to the invariant risk bound in (6), notice that it is composed of two parts, namely, the statistical risk $R$ from (SRM) and what we have shown in (8) to be a intance/model-dependent data augmentation. However, in order to address the potential biases introduced by transformations, rather than modifying the objective as in standard data augmentation, we propose to combine these two terms in a constrained learning problem. Explicitly,

$$P^\star = \min_{\boldsymbol{\theta} \in \Theta} \quad \mathbb{E}_{(\mathbf{x},y)\sim\mathfrak{D}} \left[ \ell(f_{\boldsymbol{\theta}}(\mathbf{x}), y) \right] \tag{CSRM}$$

$$\text{s. to } \mathbb{E}_{(\mathbf{x},y)\sim\mathfrak{D}} \left[ \mathbb{E}_{g \sim \lambda^\star} \left[ \ell(f_{\boldsymbol{\theta}}(g\mathbf{x}), y) \right] \right] \leq \epsilon \ .$$

Notice that this formulation tackles the two terms forming the invariant risk bound in (6), but instead of combining them directly, it incorporates the data augmentation term as a constraint in the typical statistical risk minimization problem (SRM). This formulation has the advantage that if a solution to the unconstrained problem is feasible, i.e., satisfies the invariance constraint in (CSRM), the presence of that constraint has no effect on the statistical problem. Yet, it can be beneficial when approximating the solution of (CSRM) empirically. We will explore this fact in the next section, where we tackle the practical challenges involved in solving (CSRM).

For conciseness in (CSRM) we have included only one invariance constraint associated with a single set of transformations $\mathcal{G}$. However, our approach can be extended to an arbitrary number of constraints defined by transformation sets $\mathcal{G}_i$, $i = 1, \ldots, m$ (each inducing an augmentation distribution $\lambda_i^\star$), and constraint levels $\epsilon_i$. All of the following derivations still hold, regardless of the number of constraints.

## 4 Algorithm Development

Solving (CSRM) presents two challenges. First, it is a constrained statistical learning problem, which involves the unknown data distribution $\mathfrak{D}$. We address this by resorting to an empirical dual problem as explained on Section 4.1. Second, it can be hard to sample from $\lambda^\star$. We address this by introducing a smooth approximation that leverages MCMC methods on Section 4.2.

## 4.1 Empirical Dual Constrained Learning

To tackle the invariance-constrained statistical risk minimization problem we leverage recent duality results in constrained learning theory (Chamon et al., 2021), that approximate the problem by its

empirical dual

$$D_{\text{emp}}^{\star} = \max_{\gamma \geq 0} \min_{\boldsymbol{\theta} \in \Theta} \frac{1}{n} \sum_{i=1}^{n} \ell(f_{\boldsymbol{\theta}}(\mathbf{x}_i), y_i) + \gamma \left( \frac{1}{n} \sum_{i=1}^{n} \mathbb{E}_{g \sim \lambda^{\star}} \left[ \ell(f_{\boldsymbol{\theta}}(g\mathbf{x}), y) \right] - \epsilon \right). \quad \text{(D-CERM)}$$

The advantage of (D-CERM) is that it is an unconstrained problem that, provided we have enough samples and the parametrization is rich enough, can approximate the constrained statistical problem ( CSRM). Namely, the difference between the optimal value of the empirical dual $D_{\text{emp}}^{\star}$ and the statistical primal $P^{\star}$, i.e., the empirical duality gap is bounded (Chamon et al., 2021).

As in regular data augmentation, we will also approximate the expectation over $\lambda^{\star}$ by sampling transformations as discussed in Section 4.2. Then, the problem D-CERM becomes an unconstrained deterministic problem, which can be solved using the algorithm described in Section 4.3.

Note that finding a Lagrangian minimizer for a fixed value of the dual variable ($\gamma$) is equivalent to minimising the risk under a fixed mixture augmentation distribution[1], where the value of $\gamma$ controls the probability of sampling the identity. We can also interpret this as optimising a penalised or regularised learning objective. However, solving the constrained problem, namely maximising over $\gamma$, has fundamental differences.

First, constraints explicit the requirement they represent. While the degree of invariance imposed should depend only on the statistical problem at hand, the value of $\gamma$ needed to achieve it will depend on the sample size, the parametrization and the learning algorithm. In contrast, constrained learning dynamically adjusts the amount of augmentation — dictated by $\gamma$ — to a particular learning setup.

Second, the optimal dual variable can give information about the trade-off minimising the loss over training samples and satisfying the invariance constraint. In penalised approaches, on the contrary, this trade-off is fixed.

Lastly, the aforementioned informativeness and interpretability can facilitate hyper-parameter tuning. The insights gathered from optimal dual variables can be leveraged a posteriori, for instance, to manually choose appropriate transformations, relax the invariance constraint levels, or change the learning setup (e.g. increase the capacity of the model class).

## 4.2 SAMPLING TOWARDS INVARIANCE

Sampling the augmentation distribution $\lambda^{\star}$ can be difficult when $\mathcal{G}$ is not finite and $f_{\boldsymbol{\theta}}$ is a deep neural network. Even when the transformation space is low dimensional, as in the case of translations and rotations, the highly non-convex loss landscape of these models makes the maximization over $\mathcal{G}$ challenging (Engstrom et al., 2017). If the optimal distribution $\lambda^{\star}$ is not smooth, it is challenging to sample from it with sufficient accuracy (Homem-de Mello and Bayraksan, 2014). Consequently, obtaining an unbiased estimator of $\mathbb{E}_{g \sim \lambda^{\star}} \left[ \ell(f_{\boldsymbol{\theta}}(g\mathbf{x}), y) \right]$ may not be possible. Therefore, we add an $L_2$ norm penalisation, which promotes smoothness, to leverage MCMC methods.

We then define the *c-smoothed distribution* $\lambda_c^{\star}$ as a solution to the regularised problem

$$\lambda_c^{\star} \subseteq \text{argmax}_{\lambda \in \mathcal{L}_+^2} \int_{\mathcal{G}} \lambda(g) \ell(f_{\boldsymbol{\theta}}(g\mathbf{x}), y) dg + c \int_{\mathcal{G}} \lambda(g)^2 dg,$$

$$\text{s. to} \int_{\mathcal{G}} \lambda(g) dg = 1$$

The regularization term introduces an optimality gap with respect to worst case perturbations, i.e., $\mathbb{E}_{g \sim \lambda_c^{\star}} \left[ \ell(f_{\boldsymbol{\theta}}(g\mathbf{x}), y) \right] \leq \max_{g \in \mathcal{G}} \ell(f_{\boldsymbol{\theta}}(g\mathbf{x}), y)$. However, for particular values of $c$ the regularized problem has a closed form solution (Robey et al., 2021a) that allows us to sample from it easily. Namely, there exists a constant $c \geq 0$ such that $\lambda_c^{\star}(\mathbf{x}, y, g) = \frac{\ell(f_{\boldsymbol{\theta}}(g\mathbf{x}), y)}{c}$.

Since $\lambda_c^{\star}$ is a smooth probability distribution, we do not need to estimate the multiplicative factor $c$ to sample from it by leveraging Monte Carlo Markov Chain methods (MCMC).

MCMC methods (Hastings, 1970) are based on constructing a Markov chain that has the target distribution as an equilibrium distribution. *Independent* Metropolis Hastings uses a state independent

---

[1]Explicitly, with probability $\frac{1}{1+\gamma}$, the identity is sampled, and with probability $\frac{\gamma}{1+\gamma}$, $g \sim \lambda^{\star}$.

- usually fixed - proposal for each step. In our case, it only requires applying a transformation and computing a forward pass of the neural network to evaluate the loss. This enables the use of non-differentiable transformations, and has the advantage that the density at consecutive proposals can be evaluated in parallel allowing speedups in the sampling step. Although MH methods thus allow to sample the proposal distribution with low computational cost, they exhibit random walk behaviour, which leads to slow convergence in high dimensional settings (Dellaportas and Roberts, 2003; Holden et al., 2009).

We can then approximate $\mathbb{E}_{g \sim \lambda_c^\star} [\ell(f_{\boldsymbol{\theta}}(g\mathbf{x}_i), y_i)]$ by sampling transformations according to the loss. Namely, we can obtain a set of $m$ samples drawn from $\lambda_c^\star$ and approximate the expectation over the group by the sample mean

$$
\mathbb{E}_{g \sim \lambda_c^\star} [\ell(f_{\boldsymbol{\theta}}(g\mathbf{x}_i), y_i)] \approx \frac{1}{m} \sum_{j=1}^{m} \ell(f_{\boldsymbol{\theta}}(\mathbf{g_j}\mathbf{x_i}), y_i),
$$

where $g_1, \ldots, g_m \overset{\text{i.i.d.}}{\sim} \ell(f_{\boldsymbol{\theta}}(g\mathbf{x}_i), y_i)/c$ are $m$ transformations sampled from the smoothed distribution $\lambda_c^\star(f_{\boldsymbol{\theta}}, \mathbf{x}_i, y_i)$.

In the next section, the implementation of independent-MH with a uniform proposal is described in Algorithm 2, together with the primal-dual augmentation algorithm.

### 4.3 PRIMAL-DUAL ALGORITHM

Since the cost of the inner minimization, i.e. training the model, can be high, we adopt an alternating update scheme (K. J. Arrow and Uzawa., 1958) for the primal and dual variables, as in (Chamon et al., 2021; Fioretto et al., 2020).

A bounded empirical duality gap *does not* guarantee that the primal variables obtained after running the alternating primal-dual algorithm 1 and solving the saddle point problem approximately are near optimal or approximately feasible. Although stronger *primal recovery* guarantees can be obtained by randomizing the learning algorithm (Chamon et al., 2021), it requires storing model parameters $\boldsymbol{\theta}$ at each iteration, and there is empirical evidence (Chamon et al., 2021; Robey et al., 2021a; Elenter et al., 2022; Shen et al., 2022; Cervino et al., 2022; Zhang et al., 2022) that good solutions can still be obtained without randomization.

Algorithm 2 describes transformation sampling. By keeping only one sample ($m = 1$) we recover the usual augmentation setting, that yields one augmentation per sample in the training batch. In our experiments we address this setting, since it has lower computational costs. However, simply keeping more samples from the chain ($m > 1$) allows to extend the method to the batch augmentation setting (Hoffer et al., 2020), which creates several augmented samples from the same instance in each batch.

Although several steps of the chain may be required to deviate enough from the proposal distribution, we show in the experimental section that sampling the constraint approximately by using few sampling steps suffices.

---

**Algorithm 1** Primal-Dual Augmentation

---

1: $\lambda = 0$, $\boldsymbol{\theta} = \boldsymbol{\theta}_0$.
2: **for** Batch in $\{(\mathbf{x}_i, y_i)\}_{i=1}^{n}$ **do**
3:     **for** $(\mathbf{x}_i, y_i) \in$ Batch **do**
4:         $g_{i1}, \ldots, g_{im} \sim^{iid} \ell(f_{\boldsymbol{\theta}}(g\mathbf{x}_i), y_i)/c$             $\triangleright$ Sample transformations
5:     $s = \frac{1}{|\text{Batch}|} \sum_{(\mathbf{x}_i, y_i) \in \text{Batch}} \left[ \frac{1}{m} \sum_{j=1}^{m} \ell(f_{\boldsymbol{\theta}}(g_{ij}\mathbf{x}_i), y_i) \right] - \epsilon$    $\triangleright$ Evaluate constraint slack
6:     $\ell = \frac{1}{|\text{Batch}|} \sum_{(\mathbf{x}_i, y_i) \in \text{Batch}} \ell(f_{\boldsymbol{\theta}}(\mathbf{x}_i), y_i)$               $\triangleright$ Evaluate Loss
7:     $\hat{L} = \ell + \gamma s$                             $\triangleright$ Compute Lagrangian
8:     $\boldsymbol{\theta} = \boldsymbol{\theta} - \eta_p \nabla_{\boldsymbol{\theta}} \hat{L}$                        $\triangleright$ Primal update
9:     $\gamma = [\gamma + \eta_d s]_+$                           $\triangleright$ Dual Update

---

---

**Algorithm 2** Independent MH sampler

---

1: $g^{(0)} \sim \mathcal{U}(\mathcal{G})$              ▷ Sample initial State
2: $\ell^{(0)} = \ell\left(f_{\boldsymbol{\theta}}\left(g^{(0)}\mathbf{x}\right), y\right)$           ▷ Evaluate loss
3: **for** $t = 1, \ldots, n_{steps}$ **do**
4:   $g_{\text{prop}} \sim \mathcal{U}(\mathcal{G})$          ▷ Sample next proposal
5:   $\ell_{\text{prop}} = \ell\left(f_{\theta}\left(g_{\text{prop}}\mathbf{x}\right), y\right)$       ▷ Evaluate Loss
6:   $p = \min\left(1, \frac{\ell_{\text{prop}}}{\ell^{(t-1)}}\right)$        ▷ Acceptance Prob
7:   w.p. $p$:             ▷ Accept/Reject
8:    $g^{(t)} = g_{\text{prop}}, \ell^{(t)} = \ell_{\text{prop}}$
9:   else:
10:    $g^{(t)} = g^{(t-1)}, \ell^{(t)} = \ell^{(t-1)}$

---

## 5   AUTOMATIC DATA AUGMENTATION EXPERIMENTS

This section showcases Algorithm 1 in common image classification benchmarks. We compare it to state-of-the-art data augmentation methods in terms of classification accuracy. Furthermore, we discuss other advantageous properties of our method through ablations. Namely, we focus on the ability to control the effect of data augmentation by modifying the constraint level, the informativeness of dual variables, and the benefits of adapting the augmentation distribution throughout training. We follow the setup (including the transformation sets) used in recent automatic data augmentation literature (Müller and Hutter, 2021). A complete list of transformations together with other hyperparameters and training settings can be found on Appendix B. Note that four out of the sixteen transformations used are non-differentiable. Whereas other works have introduced gradient approximations for transformation operations with discrete parameters (Li et al., 2020b; Hataya et al., 2020), our approach does not require such approximations.

Throughout these experiments, we fixed the number of steps of the MH sampler (Algorithm 2) to two, which has the added advantage of reducing the computational cost of evaluating proposals. The constraint level was determined by a grid search targeting cross-validation accuracy. As shown in Table 1, in both transformation sets considered, we find that our approach improves or closely matches existing approaches. The failure to achieve large improvements in accuracy over baselines, which has been attributed to a stagnation in data augmentation research (Müller and Hutter, 2021), can also reflect the limits of the benchmarking setup.

| | Standard | | | Wide | | |
|---|---|---|---|---|---|---|
| | TA | DeepAA | **OURS** | TA | DeepAA | **OURS** |
| CIFAR10 | | | | | | |
| Wide-ResNet-40-2 | $96.55 \pm 0.11$ | $96.43 \pm 0.09$ | $\mathbf{96.76 \pm 0.14}$ | $96.24 \pm 0.19$ | $96.27 \pm 0.19$ | $\mathbf{97.05 \pm 0.18}$ |
| Wide-ResNet-28-10 | $97.46 \pm 0.10$ | $97.57 \pm 0.15$ | $\mathbf{97.74 \pm 0.10}$ | $97.51 \pm 0.20$ | $97.27 \pm 0.10$ | $\mathbf{97.85 \pm 0.17}$ |
| CIFAR100 | | | | | | |
| Wide-ResNet-40-2 | $79.92 \pm 0.13$ | $79.45 \pm 0.42$ | $\mathbf{80.83 \pm 0.31}$ | $79.96 \pm 0.45$ | $79.36 \pm 0.27$ | $\mathbf{81.19 \pm 0.34}$ |
| Wide-ResNet-28-10 | $83.40 \pm 0.16$ | $\mathbf{83.77 \pm 0.29}$ | $83.53 \pm 0.16$ | $84.11 \pm 0.24$ | $83.09 \pm 0.30$ | $\mathbf{84.89 \pm 0.12}$ |
| SVHNcore | | | | | | |
| Wide-ResNet-28-10 | $98.05 \pm 0.03$ | $98.04 \pm 0.08$ | $\mathbf{98.15 \pm 0.09}$ | $\mathbf{98.07 \pm 0.03}$ | $97.93 \pm 0.03$ | $98.01 \pm 0.04$ |

Table 1: Image Classification accuracy for WideResnet architectures (Zagoruyko and Komodakis, 2016) trained using different augmentation policies, defined on standard (Cubuk et al., 2018) and wider (Müller and Hutter, 2021) augmentation search spaces. We include state-of-the-art methods TA (Müller and Hutter, 2021) and DeepAA (Zheng et al., 2022), and 95% confidence intervals computed over five independent runs. In CIFAR datasets our approach leads to an improvement in accuracy, whereas in SVHN (Netzer et al., 2011) it leads to a degradation in performance in the wide setup.

Regardless, our approach yields improvements in test accuracy over a baseline model without augmentation for a wide range of constraint levels (Figure 1). This illustrates the robustness of the solution to this hyperparameter. Observe also that as the constraint is relaxed (by increasing $\epsilon$), the

training error decreases while the generalization gap, i.e., the difference between train and test errors, increases. In other words, by loosening the invariance requirement the model can fit better to training samples at the cost of worse generalization. Eventually, only the generalization gap increases while the training error stagnates ($\epsilon > 2.1$ for CIFAR100 and $\epsilon > 0.8$ for CIFAR10). This transition occurs at the same point at which the final value of the dual variable $\gamma$ from (D-CERM) essentially vanishes (darker color). This showcases the inpromativeness of the dual variable. In the case of CIFAR10, even the training error begins to increase at that point, suggesting that the invariance requirement need not be at odds with accuracy.

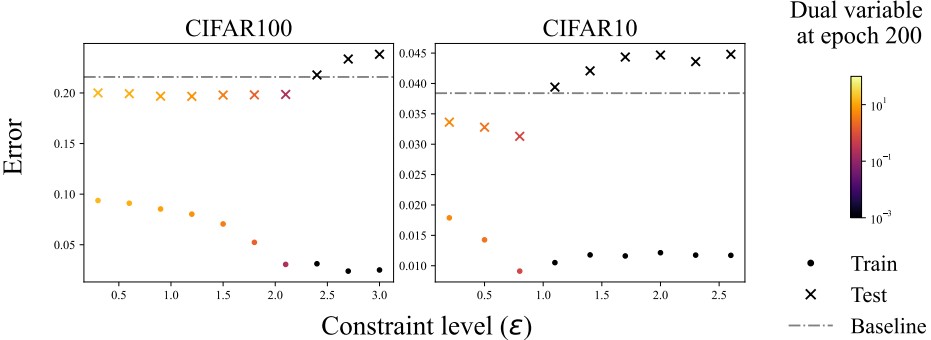

Figure 1: Constraint level ablation for WideResnet-40-2 in CIFAR datasets. We plot error rates computed over the train and test set and averaged over five runs, for different constraint levels. We include the test error of the unconstrained baseline (without augmentation) for comparison. The color of markers denotes the final value of the dual variable.

Not only does Algorithm 1 tune the effect of data augmentation on the solution (by adapting $\gamma$ in step 9), but also modifies the distribution over transformations during training (step 4). To showcase the benefits of this over the use of a fixed distribution, Figure 2 compares the results obtained using our approach (sampling according to $\lambda_c^\star$) and one where step 4 is replaced by a uniform sampling over transformations. For the same constraint levels, lower test errors are obtained by sampling transformations according to $\lambda_c^\star$, i.e., promoting invariance. Note also that for $\epsilon = 2.1$ in CIFAR100 and $\epsilon = 0.8$ for CIFAR10, the performance gap is quite large. Once again, this occurs at the point in which $\gamma$ vanishes (darker color) for the uniform distribution, i.e., no data augmentation occurs by the end of training. At this point, however, there is still value in promoting invariance by sampling from $\lambda_c^\star$ as evidenced by the positive value of the dual variable (lighter color) in this approach.

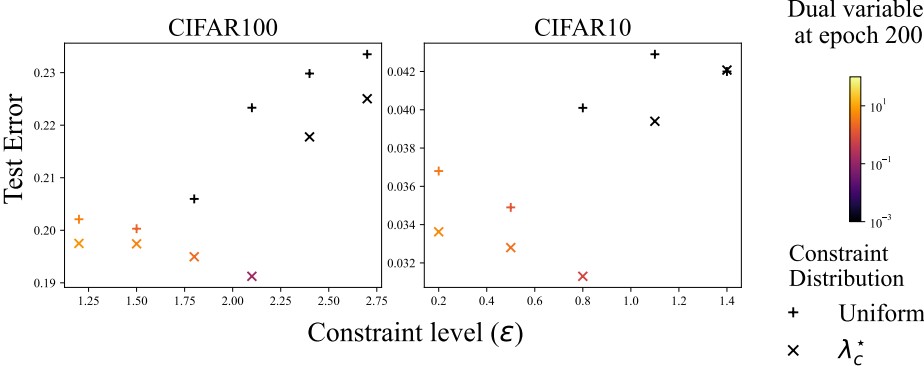

Figure 2: We compare our approach to a constraint on the uniform distribution, for WideResnet-40-2 in CIFAR datasets, at different constraint levels. We plot error rates computed over test set and averaged over five runs. Markers denote the augmentation distribution. The color of markers denotes the final value of the dual variable.

Furthermore, to assess the impact of the sampling approximation on the performance of the solution we conduct an ablation on the number of steps of the MH sampler (Algorithm 2), keeping the constraint level ($\epsilon$) fixed. Using more steps of the chain allows samples to deviate further from the uniform distribution, which as shown in Figure 3 is reflected on the lower entropy of sampled transformations (first row). As shown in the second row, this results in higher classification error rates on the training set (represented by blue circular markers). However, we find that increasing the number of steps does not affect significantly test error (denoted by orange crosses). In Appendix C.1.3 we show how the number of sampling steps affects the evolution of dual variables.

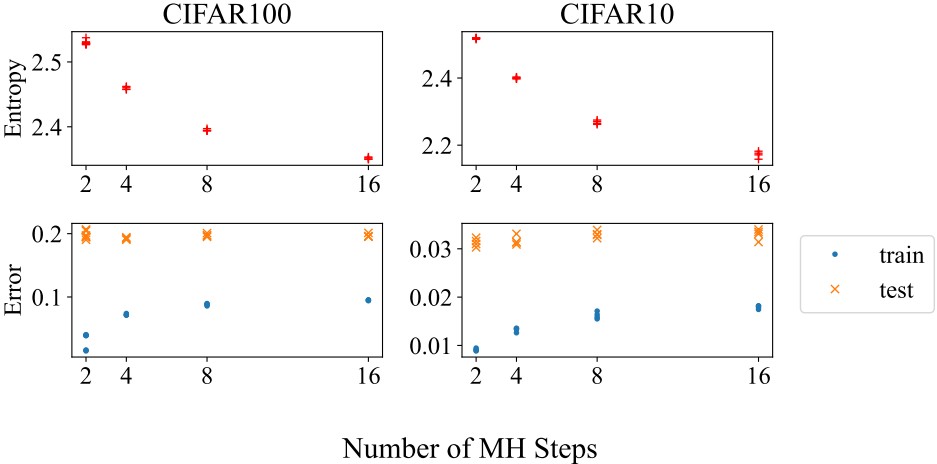

Figure 3: Number of Metropolis Hastings steps ablation for WideResnet-40-2 in CIFAR datasets. The constraint level is fixed for each dataset (0.8 in CIFAR10 and 2.1 for CIFAR100). The first row shows the entropy of the augmentations sampled at the last epoch of training. The second rows classification error rates. Each point represents an independent run, experiments are repreated four times.

## 6 CONCLUSION

In this paper, we have proposed a constrained learning approach for automatic data augmentation, which instead of using augmented samples as a modified learning objective, imposes an invariance-constraint. We have shown that this yields an augmentation distribution that adapts during training, and found that coarse sampling approximations based on MCMC methods can improve generalization in automatic data augmentation benchmarks. Furthermore, our experiments showed that the dual variable can give insights about the resulting augmentation distribution. We also found that strictly feasible solutions were obtained for a wide range of constraint levels, with notably different generalization gaps, and that in some cases tightening the constraint even led to a lower training error. Thus, analysing the interplay between the learning problem and the optimization algorithm is a promising future work direction.

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

# A    ADDITIONAL RELATED WORK

## A.1    CONSTRAINED LEARNING AND DATA AUGMENTATION

Xu et al. (2021) also formulate data augmentation as a constrained learning problem. They impose a constraint on the excess risk, i.e. the difference between the statistical risk and its optimal value, on augmented data. Thus the constraint level on the augmented risk is also determined by the data distribution, augmentations considered, and model class, and the existence of a strictly feasible point is guaranteed.

$$\min_{\boldsymbol{\theta} \in \Theta} R(f_{\boldsymbol{\theta}}) \quad \text{s.t.} \quad R_{aug}(f_{\boldsymbol{\theta}}) - \min_{\hat{\boldsymbol{\theta}} \in \Theta} R_{aug}(f_{\hat{\boldsymbol{\theta}}}) \le \epsilon, \quad \epsilon > 0,$$

where $R$ and $R_{aug}$ are the statistical risk under the original and augmented distribution, exlipicity

$$R(f_{\boldsymbol{\theta}}) = \mathbb{E}_{(\mathbf{x},y) \sim \mathfrak{D}}[\ell(f_{\boldsymbol{\theta}}(\mathbf{x}), y)], \quad R_{\text{aug}}(f_{\boldsymbol{\theta}}) = \mathbb{E}_{\substack{(\mathbf{x},y) \sim \mathfrak{D} \\ g \sim \mathfrak{G}}}[\ell(f_{\boldsymbol{\theta}}(g\mathbf{x}), y)].$$

Unlike our formulation, this formulation assumes a fixed distribution of augmentations $\mathfrak{G}$ is given.

By formulating it as a constrained problem, they aim to avoid introducing a bias when the data distribution is not invariant to augmentations. Two types of biases induced by augmentation are explicitly addressed, covariate shift (i.e. label-preserving augmentations) and concept shift (i.e. label mixing augmentations).

Interestingly, they show that under some conditions on the risk, augmented risk and constraint level, by utilizing the augmented data to constrain the solution to a small region SGD can achieve lower error (Xu et al., 2021, Proposition 1).

Instead of resorting to constrained optimization algorithms, they propose a two stage algorithm that consists of first finding an approximate minimizer of the augmented risk and then using that solution as an initialisation to the (unconstrained) statistical risk minimization problem. The first stage obtains a feasible point, and then under some conditions the SGD iterates obtained when solving the second problem remain feasible (Xu et al., 2021, Theorem 1).

## A.2    ADVERSARIAL DATA AUGMENTATION

Zhang et al. (2020) have shown that using adversarial transformations - which as already mentioned is related to promoting invariance - can give competitive results with respect to other automatic augmentation methods in image classification. However, Blaas et al. (2021) have since evidenced the importance of two factors: the implicit learning curricula and the suboptimality of the adversarial used by Zhang et al. (2020), which mitigates the biases introduced by worst-case transformations. Furthermore, Blaas et al. (2021) report that an explicit cyclic curricula in which augmentations are *mild* at first, then get harder as training progresses, and finally revert to milder augmentations at the end of training, performs better empirically. We note some interesting commonalities with our approach and experimental results. First, the dynamics of our primal-dual algorithm resemble the aforementioned heuristically defined curricula. Second, the suboptimality with respect to worst case perturbations can be related to the smoothed approximation used in our approach.

## A.3    CONSTRAINED LEARNING AND DOMAIN GENERALIZATION

Domain Generalization (DG) involves training the model in different but related data distributions, and evaluating in an unseen *domain*. For example, a common benchmark consists of domains created by rotating images in the MNIST dataset by different angles. Constrained formulations have been proposed in this context (Robey et al., 2021b; Zhang et al., 2022), that enforce invariance under learnt *domain translation* transformations. In contrast, our approach addresses pre-defined transformations that are commonly used in data augmentation pipelines in order to improve generalization in the same domain used to train the model.

## A.4    GROUP INVARIANCE

The relationship between convolutional structure and equivariance has been long known in algebraic signal processing theory (Püschel and Moura, 2006). Recently, necessity results have re-gained

attention in the context of neural network architecture design (Kondor and Trivedi, 2018). Several Group Convolutional neural network architectures that generalize CNNs to different groups by leveraging group convolutions have been proposed (Cohen and Welling, 2016; Esteves et al., 2017; Finzi et al., 2020). In the case of images, it has been shown that that SE2 equivariant layers can be implemented efficiently using regular 2D convolutions (Weiler and Cesa, 2019). This approach allows to derive a parametrization for CNN filters under finite subgroups of SE2. Among other works (Bietti and Mairal, 2019; Sannai et al., 2019) give theoretical analyses of the benefits of group invariance in learning settings.

As already mentioned, achieving invariance through architecture design is both challenging and limited in the sense that it relies on transformations having a specific structure (e.g: a group). The goal of exact invariance over the whole input space is more strict than the approximate invariance notion that our work addresses. As argued by Mallat (2016), CNNs can learn *locally* invariant features with respect to arbitrary transformation groups, which could explain their generalization properties. Furthermore, empirical studies evidence modern CNN architectures learn approximately equivariant features to transformations such as scaling and rotations (Olah et al., 2020), or diffeomorphisms (Petrini et al., 2021), even when trained without direct augmentation.

However, commonly used augmentations do not form a group. Our approach does not require this structure.

## B    EXPERIMENTAL SETUP DETAILS

### B.1    TRANSFORMATIONS

As in recent automatic augmentation literature (Ho et al., 2019; Lim et al., 2019; Hataya et al., 2020; Zhang et al., 2020; Cubuk et al., 2020; LingChen et al., 2020; Zhang et al., 2020; Müller and Hutter, 2021; Zhou et al., 2021b; Zheng et al., 2022; Cheung and Yeung, 2022), we focus on image classification datasets and employ a transformation search space comprising 14 operations, introduced by Cubuk et al. (2018). For those that have parameters, their magnitudes are discretized in thirty levels, which does not compromise performance and greatly reduces the search space (Cubuk et al., 2020). Most approaches compose transformations, i.e. applying more than one transformation to the same image. However, recently Müller and Hutter (2021) have shown that applying only one transformation at a time, defined over a wider magnitude space (noted Wide in Table 2) can outperform other approaches. We thus use the same transformation space as (Müller and Hutter, 2021).

Table 2 from Müller and Hutter (2021) lists the operations and their magnitude ranges. In our experiments we used the wide search space, the standard ranges from Cubuk et al. (2020) are included for comparison. We extend the codebase provided by Müller and Hutter (2021), which uses the Pillow[2] implementation of all transformations except for cutout, and refer to its documentation for further details about image operations.

| Operation | Magnitude | |
|---|---|---|
| | Standard | Wide |
| Identity | — | — |
| ShearX | $[-0.3, 0.3]$ | $[-0.99, 0.99]$ |
| ShearY | $[-0.3, 0.3]$ | $[-0.99, 0.99]$ |
| TranslateX | $[-10, 10]$ | $[-32, 32]$ |
| TranslateY | $[-0.45, 0.45]$ | $[-32, 32]$ |
| Rotate | $[-30, 30]$ | $[-135, 135]$ |
| AutoContrast | — | — |
| Invert | — | — |
| Equalize | — | — |
| Solarize | $[0, 256]$ | |
| Posterize | $[4, 8]$ | $[2, 8]$ |
| Contrast | $[0.1, 1.9]$ | $[0.01, 2]$ |
| Color | $[0.1, 1.9]$ | $[0.01, 2]$ |
| Brightness | $[0.1, 1.9]$ | $[0.01, 2]$ |
| Sharpness | $[0.1, 1.9]$ | $[0.01, 2]$ |
| Flips | — | — |
| Cutout | $16(60)$ | $16(60)$ |
| Crop | — | — |

Table 2: Pillow image operations in the data augmentation search space and the range of magnitudes corresponding to the standard (Cubuk et al., 2020) and wide (Müller and Hutter, 2021) search spaces. Some operations do not use magnitude parameters.

### B.2    TRAINING SETUP

In order to enable comparisons and reproducibility we use the same training pipeline as in previous works Müller and Hutter (2021) . We apply the vertical flip and the pad-and-crop augmentations

---

[2] https://github.com/python-pillow/Pillow

and a 16 pixel cutout (DeVries and Taylor, 2017) after any augmentation method. We trained Wide-ResNet (Zagoruyko and Komodakis, 2016) models in the 40-2 and 28-10 configurations.

Except for epoch ablation experiments, we use SGD with Nesterov Momentum and a learning rate of 0.1, a batch size of 128, a 5e-4 weight decay, and cosine learning rate decay. In ablation experiments we also trained for 600 epochs and used a custom learning rate schedule. For the first 185 epochs we followed the same cosine learning rate decay schedule, and then switching to a custom step learning rate scheduler detailed on Table 3. This schedule was implemented after observing that just scaling the cosine learning rate schedule to 600 epochs resulted in slow convergence, thus yielding solutions similar to training for 600 epochs, and failing reflect the effects of early stopping which this experiment addressed.

We used 2 MH steps for the smoothed adversarial unless stated otherwise, and a learning rate of $10^{-3}$ for the dual ascent step.

No other hyperparameters were tuned or modified with respect to standard settings. The constraint levels were set for Wide-ResNet-40-2 in the wide augmentation space by maximising three fold cross validation over the grid specified on Table 4. Then the constraint levels were adjusted for other architectures and search spaces so that the dual variables at the end of training were small but not zero (of the order of $10^{-1}$), which empirically showed good results. The resulting constraint levels corresponding to the results in 1 are detailed in table 5.

We used the provided code and reported hyperparameters for running TrivialAugment (Müller and Hutter, 2021) and DeepAA (Zheng et al., 2022). For the latter, unlike the results reported in (Zheng et al., 2022), we kept all evaluation hyperparameters (including weight decay) consistent with that of other methods. Results for the wide augmentation space were not included in (Zheng et al., 2022). We thus performed the augmentation policy search for the wide search space using the same hyperparameters (except for the augmentation space) reported in (Zheng et al., 2022) for CIFAR datasets. We also run the search for both augmentation spaces in SVHN datasets, and evaluated the policy with the same setup as other methods.

| Epochs | LR Scheduler Step |
|---------|-------------------|
| 180-230 | 10 |
| 230-430 | 20 |
| 430-600 | 40 |

Table 3: Learning Rate Custom schedule used when training for 600 epochs. We use the standard scheduler for the first 180 epochs, and then update the LR only every $n$ epochs, where $n$ is the number indicated in the second column. This hand designed schedule outperforms using a cosine learning rate schedule for 600 epochs, but could improvements convergence speed and performance by exploring other LR-schedulers or tuning it.

| Dataset | Constraint Level Grid Range |
|---------|------------------------------|
| CIFAR10 | [0.2, 2.3] |
| CIFAR100 | [0.3, 2.7] |
| SVHNcore | [0.1, 1.5] |

Table 4: Constraint level grid search space. For each dataset, we evaluated three fold cross validation accuracy for 8 evenly spaced constraint levels in the ranges given in the second column. The one with the highest cross validation score was then selected and used to train the model with the full dataset.

| | Constraint Level | |
|---|---|---|
| | Standard | Wide |
| CIFAR10 | | |
| Wide-ResNet-40-2 | 0.6 | 0.8 |
| Wide-ResNet-28-10 | 0.4 | 0.8 |
| CIFAR100 | | |
| Wide-ResNet-40-2 | 0.9 | 0.8 |
| Wide-ResNet-28-10 | 0.9 | 1.2 |
| SVHNcore | | |
| Wide-ResNet-28-10 | 0.1 | 0.2 |

Table 5: Constraint levels for different datasets and architectures, for the results presented in Table 1

# C   ADDITIONAL EXPERIMENTS

## C.1   AUTOMATIC DATA AUGMENTATION

The following section contains additional ablations and discussions about our algorithm. As in previous sections, we use the wide Müller and Hutter (2021) augmentation space and CIFAR image classification benchmarks. Our main motivation is to analyse how the different hyperparameter choices and the learning algorithm affect the generalization and invariance of the obtained solutions. First, we show how the dual variables adapts to different constraint levels during training, and link its dynamics to heuristically defined learning curricula. We then evaluate the effect of training for more epochs and link our observations to the known properties of early stopping in unconstrained learning, in section C.1.2. In section C.1.3, we analyse how the sampling approximation affects regularisation, by performing an ablation on the number of MH steps. Lastly, we include the observed frequencies of sampled transformations for different setups, so as to obtain further insights in how the distribution adapts throughout training (Section C.1.4).

## C.1.1   DUAL VARIABLE DYNAMICS

As already mentioned, the dual variables control the weight of augmented samples during training, thus balancing the trade-off between fitting the primal objective (i.e. loss over training samples) and satisfying the constraints (i.e. loss over transformed samples). In penalised approaches, on the contrary, this trade-off is fixed. In Figure 4 we show how the dual variable adapts to different constraint levels, for Wide-ResNet-40-2 in CIFAR datasets using the standard setup. Note that for stricter constraint levels, the algorithm has not converged when it reaches 200 epochs. We also observe that the dynamics of our primal-dual algorithm resembles the augmentation learning curricula proposed by Blaas et al. (2021).

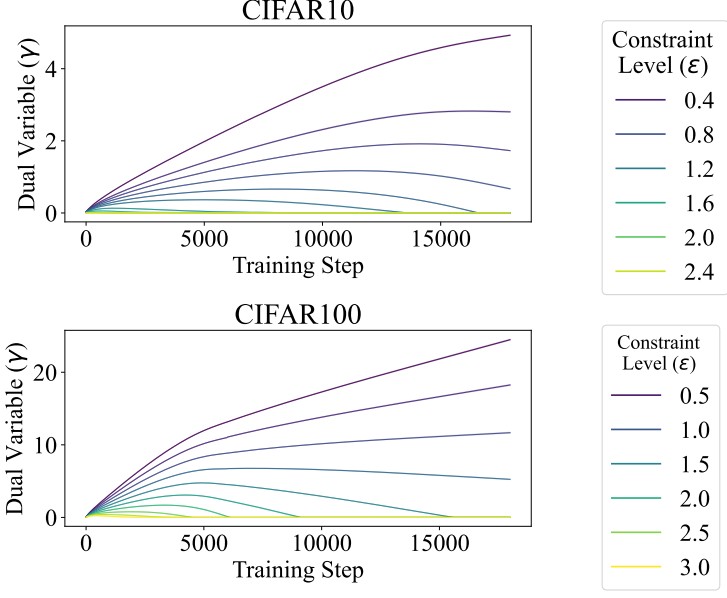

Figure 4: Evolution of dual variables during training for Wide-ResNet-40-2 in CIFAR datasets. For most levels, augmentation increases until the constraint becomes feasible and then decreases towards the end of training. For stricter levels, the algorithm does not converge in 200 epochs using the standard learning settings. However, the solutions obtained still show good properties.

## C.1.2   EARLY STOPPING

Our approach has slower convergence than unconstrained approaches and stopping training arbitrarily - after a fixed number of epochs - can result in solutions that are unfeasible or sub-optimal. However,

early stopping is a popular regulariser, particularly for neural networks trained through gradient descent. Several empirical (Caruana et al., 2000; Rice et al., 2020) and theoretical (Ji et al., 2021; Li et al., 2020a; Duvenaud et al., 2016) results show its advantages in terms of generalisation and robustness to noisy training data. In general, the advantages of early stopping do not lie in the sub-optimality of the solution in terms of training error, but on nature of the regularization or prior imposed, which leads to non-uniform model selection among models with a given training error (Cataltepe et al., 1999).

To the best of our knowledge, there is no literature that explicitly addresses early stopping in empirical primal-dual learning. Whether the generalisation gap in constraint satisfaction can be reduced by early stopping regularisation, in the same manner early stopping regularisation can reduce the generalisation gap in unconstrained learning, thus remains unclear.

Figure 5 shows an ablation on the number of epochs. Non-zero dual variables and strict feasiblility show the algorithm has not yet converged at 200 epochs. At 600 epochs whereas constraint satisfaction shows little change, training loss decreases and the generalization gap increases. Thus, we observe early stopping has a larger impact on the primal objective than on the constraint. that although for stricter levels of the constraint the algorithm has not converged when training is stopped at 200 epochs, it can yield solutions that are still feasible and have a smaller generalization gap.

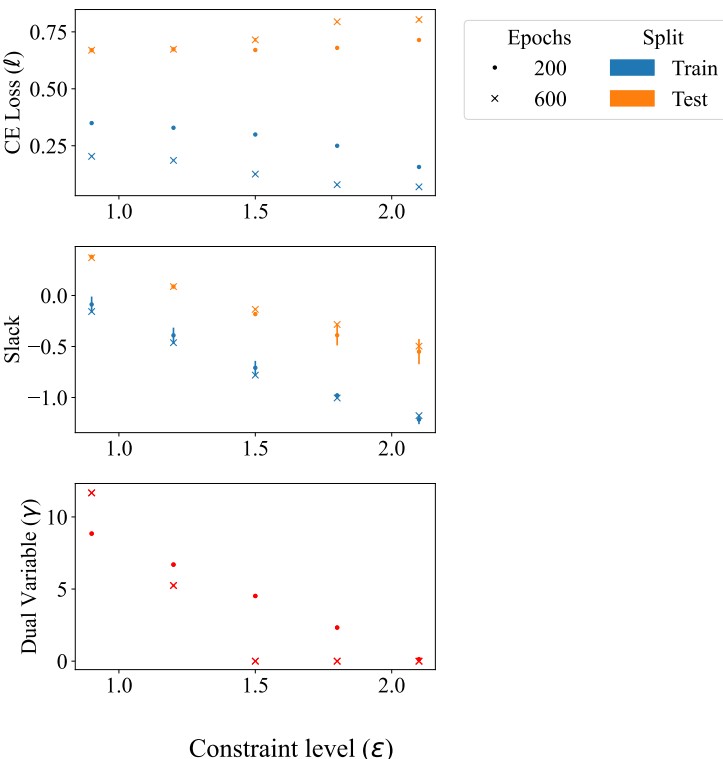

Figure 5: Training WideResnet-40-2 for 200 (standard) and 600 epochs in the CIFAR100 dataset, with different constraint levels.

### C.1.3 SAMPLING STEPS ABLATION

As already mentioned in Section 4.3, we used the Metropolis-Hastings algorithm with independent uniform proposals. As shown in Figure 6, using more steps of the chain allows samples to deviate further from the uniform distribution, as measured by the decrease in entropy. As already mentioned, dual variables give information of the trade-off between fitting clean and augmented data. We observe the final value of the dual variable is highly correlated with the entropy of sampled transformations. That is, transformations that deviate further from uniformity result in larger dual variables at the end of training. Furthermore, in 7 we show the evolution of dual variables for different sampling

steps. Sampling distributions that are closer to worst case perturbations results in more stringent requirements, and thus dual variables grow more rapidly.

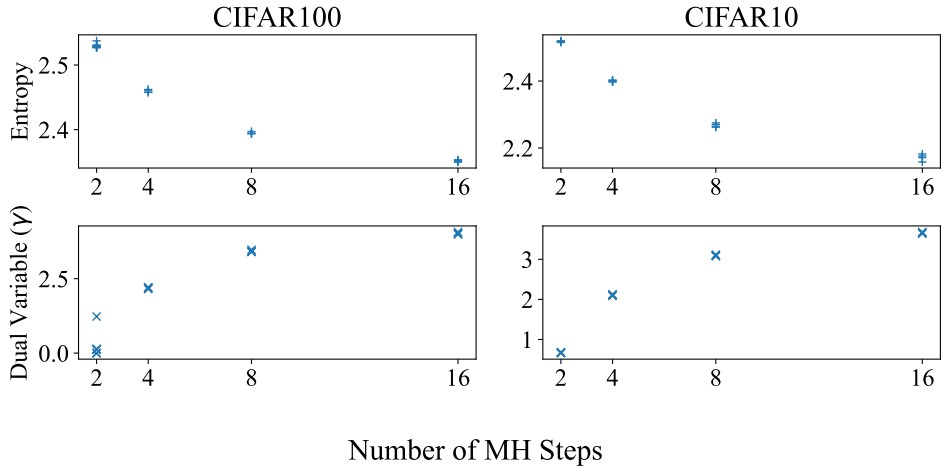

Figure 6: Number of Metropolis Hastings steps ablation for WideResnet-40-2 in CIFAR datasets. The constraint level is fixed for each dataset (0.8 in CIFAR10 and 2.1 for CIFAR100). We compute the entropy of the augmentations sampled at the last epoch of training. The value of the dual variable at epoch 200 increases with the number of steps, whereas the augmentation distribution entropy decreases. Each point represents an independent run. Experiments were repeated four times.

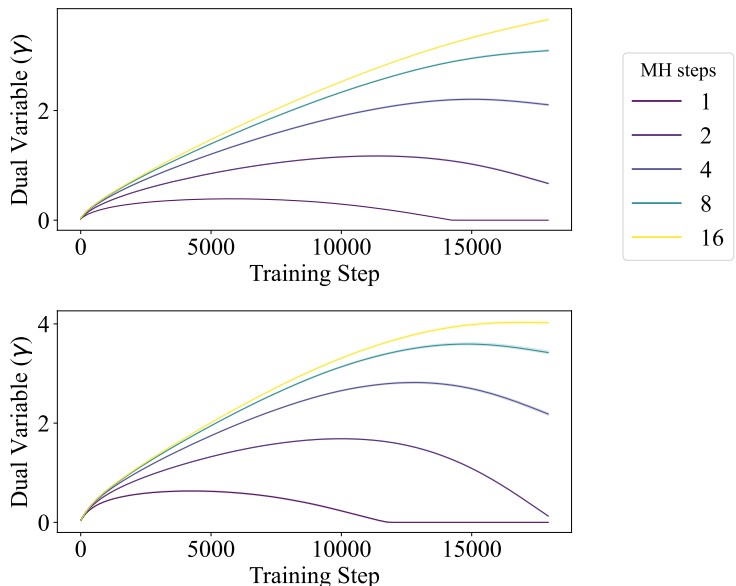

Figure 7: Evolution of dual variables during training for Wide-ResNet-40-2 in CIFAR datasets. The constraint level is fixed for each dataset (0.8 in CIFAR10 and 2.1 for CIFAR100). As the number of MH sampler steps increases so does the growth of dual variables, which reflects harder to satisfy constraints.

### C.1.4    SAMPLED TRANSFORMATIONS

We now include plots of the empirical transformation distributions. As shown in Figure 8, the frequency with which transformations are sampled varies depending on the dataset, which is a desirable property. Similar to Zhang et al. (2020), we observe a prevalence of geometric transformations,

unlike Cubuk et al. (2018). In SVHN color transformations are less frequent than in CIFAR datasets, and the frequency of geometric transformations increases, which is interesting due to the perceptual importance of shape in the digit recognition task.

As already noted, using more steps allows the chain to deviate further from the uniform proposal distribution. We thus plot the frequency of sampled transformations across training epochs on Figure 9. We observe that as training progresses, the observed frequencies also deviate further from uniformity. This suggests that, due to the dataset and architecture, some transformations may be harder to fit than others.

Similarly, we include histograms for the sampled transformation levels in Figure 10. Extreme levels are sampled more frequently, but the empirical distributions vary depending on the transformation. The deviation from uniformity and the differences between transformations are accentuated as sampling steps increase (Figure 11).

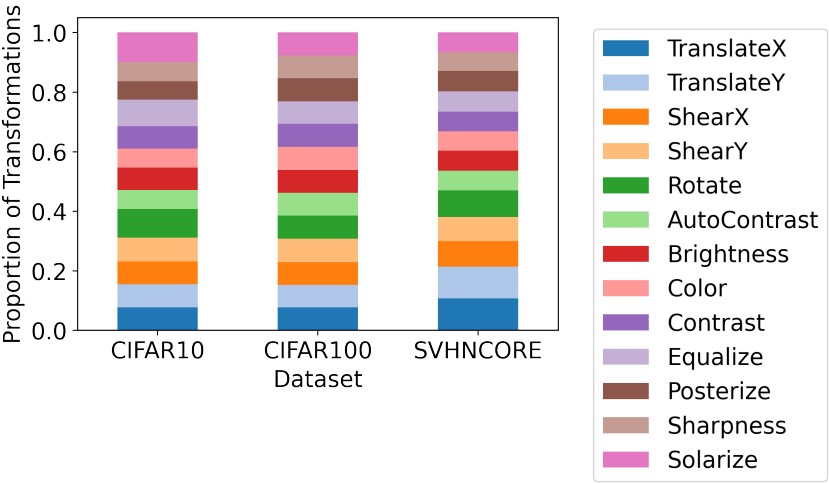

Figure 8: Frequency of sampled transformations for CIFAR and SVHN datasets using two MH sampler steps, for Wide-Resnet-28-10.

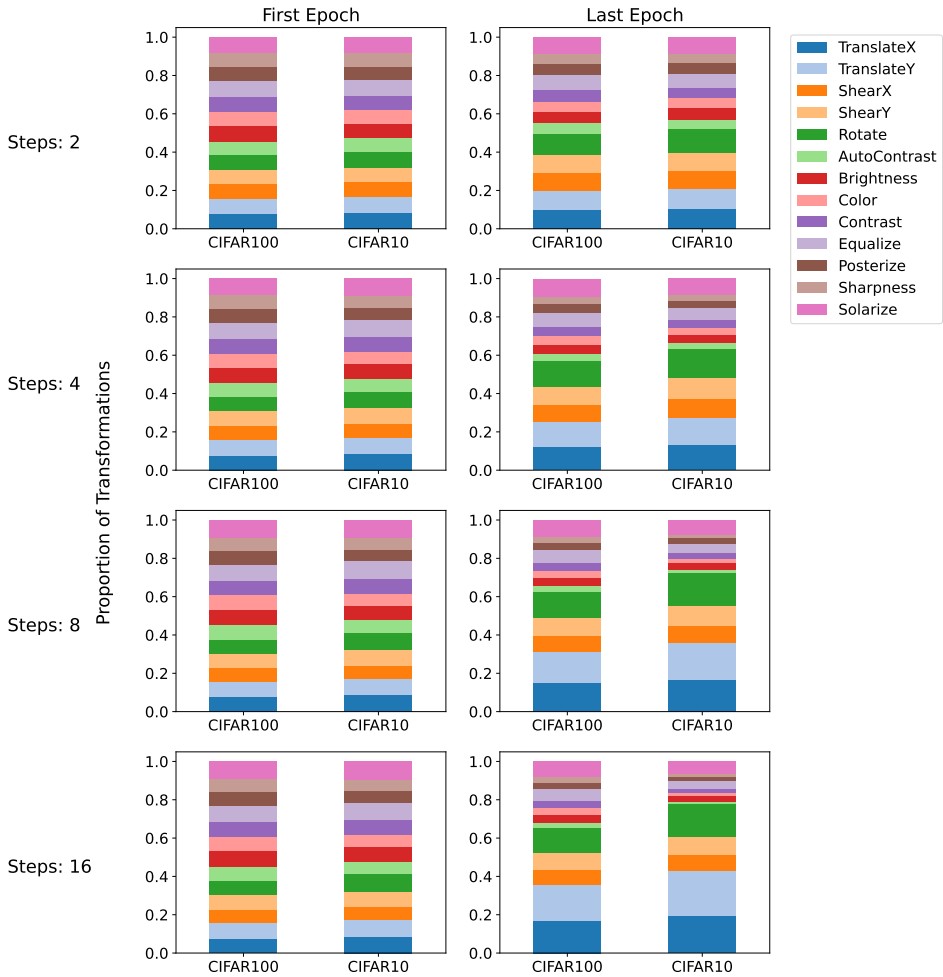

Figure 9: Frequency of sampled transformations for CIFAR datasets in the first and last epochs of training, for Wide-Resnet-40-2. As the number of steps increases, the the entropy of sampled transformations decreases, i.e., observed frequencies get further from uniformity.

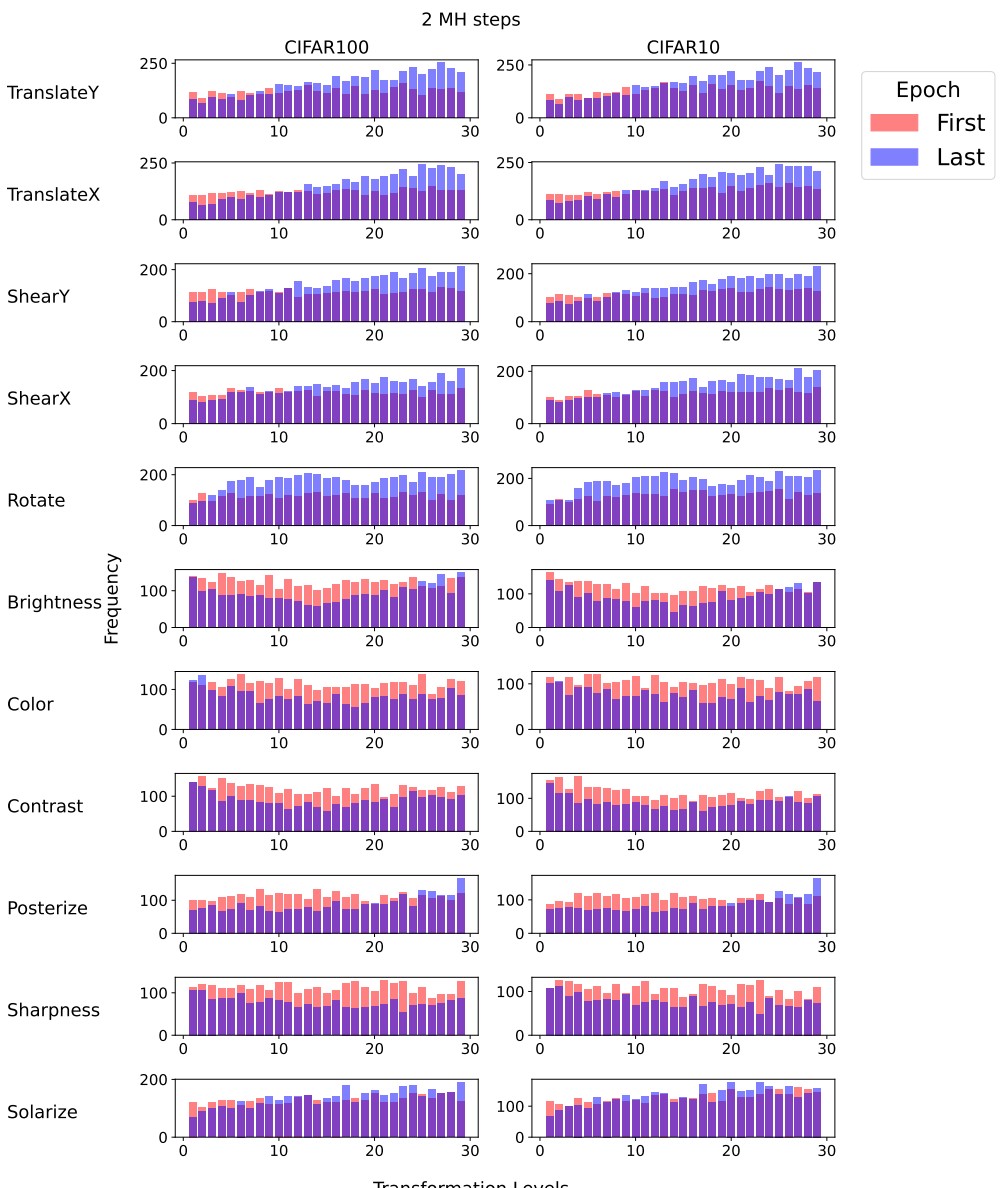

Figure 10: Frequency of sampled transformation levels across epochs, for different transformations, using two MH steps. Extreme levels are sampled more frequently, but some transformations deviate further from uniformity.

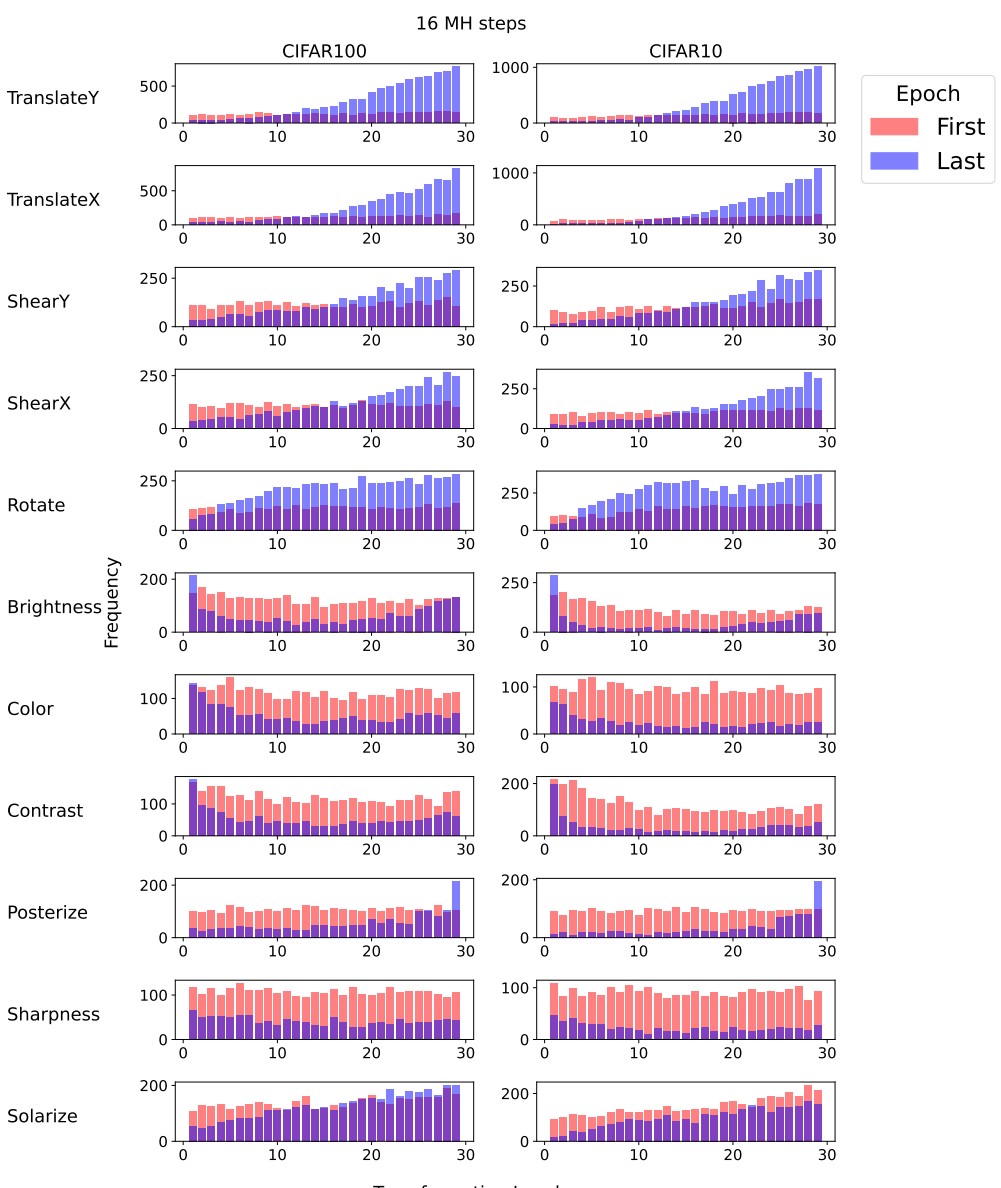

Figure 11: Frequency of sampled transformation levels for the first and last epochs, for different transformations, using sixteen MH steps. The frequencies concentrate in extreme values for some transformations.

### C.1.5 RUNTIME ANALYSIS

As mentioned in section 4.3, the added computational cost of our algorithm is that of computing a forward pass for each MCMC step. As a result, trade-off between sampling $\lambda_c^\star$ accurately and computation arises. In Table 6 we provide an empirical runtime analysis for our method for different numbers of MH steps, and compare it with the training time of baseline methods. These times correspond solely to training, and it is difficult to account for the time taken to tune the hyperparameters of each method, which hinders direct comparisons. In the case of DeepAA (Zheng et al., 2022), it requires running a data augmentation policy search that takes 11 hours (more than $5\times$ training time) using the same hardware.

| Method | | Ours | | | | |
|---|---|---|---|---|---|---|
| | TA | DeepAA | 2 steps | 4 steps | 8 steps | 16 steps |
| Epoch time (s) | 12.6 | 13.3 | 32.5 | 51.2 | 89.4 | 165.2 |

Table 6: Time per epoch for WideResnet 40-2 in CIFAR 10 dataset, on a workstation with one NVIDIA RTX 3090 GPU and AMD Threadripper 3960X (24 cores, 3.80 GHz) CPU.

### C.2 DATA AUGMENTATION CAN HINDER PERFORMANCE

As already mentioned in section A.2 there is empirical evidence that certain distributions over commonly used transformations can introduce biases that are detrimental to model performance and generalization (Blaas et al., 2021). We provide a simple experiment to show that there exist transformation distributions that make data augmentation deteriorate model performance, and that in such cases balancing the amount of data augmentation (e.g. by including the original data) is important to mitigate and overcome this issue.

We restrict the transformations in the wide augmentation space (Müller and Hutter, 2021) to their maximum magnitude. We sample transformations according to $\lambda_c^\star$ using two MH steps as previously described. We compare against training without augmentation, and training using both augmentation and the original data (i.e. adding the identity) equally weighted. While sampling transformations according to $\lambda_c^\star$ deteriorates performance with respect to the model without augmentation, including both the identity and the augmented samples achieves a superior performance, as shown in Table 7.

| No augmentation | $\lambda_c^\star$ | $\lambda_c^\star$ + Identity |
|---|---|---|
| $78.42 \pm 0.31$ | $75.19 \pm 0.54$ | $80.01 \pm 0.26$ |

Table 7: Image Classification test accuracy for WideResnet 40-2 in CIFAR100, trained using different augmentation policies defined over the wide (Müller and Hutter, 2021) augmentation space. The first column corresponds to using the training data without applying any transformations, and the second column to sampling transformations according to $\lambda_c^\star$, which results in lower accuracy. The third column corresponds to using both the original and augmented data equally weighted, which leads to an improvement in accuracy. We report the mean and confidence intervals computed over five independent runs.

### C.3 SYNTHETIC INVARIANCES

Although our approach does not aim to learn the set of symmetries or invariant transformations from the data, but rather to impose it on the predictor, dual variables can be used to gather insights on the actual invariances underlying a learning task.

We showcase this on datasets with artificial invariances, following the setup of Immer et al. (2022). Namely, we apply transformations independently drawn from the distributions specified in Table 8, to each sample in the datasets MNIST (LeCun et al., 2010) and FashionMNIST (Xiao et al., 2017). We use the same MLP and CNN architectures and hyperparameters as (Immer et al., 2022).

We run our algorithm constraining the loss on the transformation spaces detailed in Table 9, which (except for the fully rotated dataset) are larger than the true transformation range used to construct

the synthetic dataset. Note that we use the same transformation sets and constraint levels ($\epsilon$) for all synthetic datasets. As shown in table 10, except for scalings in FashionMNIST, the dual variables ($\gamma$) associated with transformations corresponding to the true synthetic invariances in the dataset are considerably smaller. This indicates that when the transformations in the constraint correspond to a true invariance of the dataset, the constraint is easier to satisfy. Since the transformation ranges in the constraint and those used to construct the dataset do not match exactly (except for the fully rotated dataset), the degree of invariance to the transformation sets in the constraints is only approximate.

| Synthetic invariance | Parameter | Distribution |
|---|---|---|
| Full Rotation | Angle in radians. | $\mathcal{U}\left[-\frac{\pi}{2}, \frac{\pi}{2}\right]$ |
| Partial Rotation | Angle in radians. | $\mathcal{U}[-\pi, \pi]$ |
| Translation | Translation in pixels. | $\mathcal{U}[-8, 8]^2$ |
| Scale | Exponential Scaling factor. | $\mathcal{U}[-log(2), log(2)]$ |

Table 8: Sampling parameters for transformations used to obtain synthetically invariant datasets, from (Immer et al., 2022).

| Constraint Set | Parameter | Range |
|---|---|---|
| Rotations | Angle in radians. | $[-\pi, \pi]$ |
| Translation | Translation in pixels. | $[-16, 16]^2$ |
| Scale | Exponential Scaling factor. | $[-1.5, 1.5]$ |

Table 9: Transformation sets used as invariance constraints. All sets are used simultanously, with the same constraint level ($\epsilon$) for all datasets.

| Dataset | Architecture | $\gamma$ | Synthetic Invariance | | | |
|---|---|---|---|---|---|---|
| | | | Full Rot. | Partial Rot. | Translation | Scale |
| MNIST | MLP | Rotation | **0.000** | **0.004** | 3.224 | 0.035 |
| | | Translation | 1.344 | 0.038 | **0.289** | 0.032 |
| | | Scale | 1.800 | 0.045 | 4.206 | **0.004** |
| | CNN | Rotation | **0.000** | **0.002** | 2.724 | 0.012 |
| | | Translation | 1.218 | 0.009 | **0.439** | 0.006 |
| | | Scale | 2.026 | 0.049 | 4.029 | **0.003** |
| F-MNIST | MLP | Rotation | **0.000** | **0.037** | 4.470 | 1.599 |
| | | Translation | 3.572 | 1.934 | **0.939** | **0.717** |
| | | Scale | 4.144 | 2.653 | 3.472 | 0.754 |
| | CNN | Rotation | **0.000** | **0.107** | 3.301 | 1.352 |
| | | Translation | 3.572 | 1.426 | **0.515** | **0.441** |
| | | Scale | 4.144 | 2.332 | 2.725 | 0.904 |

Table 10: Value of dual variables (after 400 epochs) for different transformation constraints and synthetic invariant datasets. Columns correspond to different transformations of the dataset, and rows to dual variables associated with different transformations. Except for scaling in FashionMNIST, for all architectures and dataset the dual variable associated with the constraint corresponding to the transformations applied to the dataset is considerably lower. The smallest dual variable for each dataset and architecture is bolded.

# D    RELATED NOTIONS OF INVARIANCE

There are several definitions of invariance that capture different properties of the solution or data distribution under the action of transformations in $\mathcal{G}$.

In the context of supervised learning, the data distribution is said to be *exactly invariant* iff it does not change when transformations are applied to the covariates, i.e.,

$$(\mathbf{x}, y) =_d (g\mathbf{x}, y), \text{ for all } g \in \mathcal{G},$$

where $=_d$ denotes equality in distribution. Note that this can be relaxed to approximate equality in distribution, i.e. that the distribution shift induced is small in an appropriate metric, e.g. Wassertain distance (Chen et al., 2019). This definition implies that the probability of a given label is - exactly or approximately - the same for a feature vector $\mathbf{x}$ and its transformed versions $g\mathbf{x}$:

$$P(y|\mathbf{x}) = P(y|\mathbf{gx})$$

On the other hand, a predictor can satisfy hard invariance, explicitly,

$$f_{\boldsymbol{\theta}}(\mathbf{x}) = f_{\boldsymbol{\theta}}(g\mathbf{x}) \text{ for all } g \in \mathcal{G}, \mathbf{x} \in \mathcal{X},$$

which needs to hold point-wise over the transformation set and input space. This can be relaxed by taking the mean over the data distribution to obtain a definition of *statistical invariance*, i.e.,

$$\mathbb{E}_{(\mathbf{x},y)\sim\mathfrak{D}}\left[f_{\boldsymbol{\theta}}(\mathbf{x})\right] = \mathbb{E}_{(\mathbf{x},y)\sim\mathfrak{D}}\left[f_{\boldsymbol{\theta}}(g\mathbf{x})\right] \text{ for all } g \in \mathcal{G},$$

which still needs to hold for every transformation in $\mathcal{G}$.

This notions of invariance do not explicitly contemplate the task at hand, that is, not all changes in $y$ equally affect performance. Thus, we use the loss to encode meaningful differences in labels with respect to the task, as described in section 3.1. Throughout this work, we thus refer to the invariant risk $R_{\mathrm{inv}}$ defined on equation 5 as the degree of invariance, unless otherwise noted.

