# OpenReview forum: "Automatic Data Augmentation via Invariance-Constrained Learning"
_ICLR.cc/2023/Conference — Submitted to ICLR 2023_

### Official Review · Reviewer_159N · 2022-10-23

**Confidence:** 3
**Correctness:** 3
**Technical Novelty And Significance:** 2
**Empirical Novelty And Significance:** 1
**Recommendation:** 3

**Clarity, Quality, Novelty And Reproducibility:**

- clarity:
    - good
- quality:
    - the paper needs further development on its empirical end to be considered for publication.
    - the MCMC based method can probably be further improved in terms of efficiency and performances with more modern techniques.
- novelty:
    - another highly relevant work regarding section 3
        - Toward Learning Robust and Invariant Representations with Alignment Regularization and Data Augmentation
    - Algorithms 1 and 2 seem fairly standard to me, not sure where the novelty is from.
- reproducibility:
    - seems good.

**Strength And Weaknesses:**

- strength
    - the study of generating augmented samples for invariance seems very novel to me.
    - the writing of the paper is reasonably good
- weakness
    - the algorithm still needs further development, it's hard for me to believe a MCMC based method is the best solution in this scenario, given the rich techniques developed in deep learning regime.
    - the empirical scope is quite limited: the deepAA method is barely compared.

**Summary Of The Paper:**

The paper studies an interesting problem on how to automatically generate augmented data for learning invairance representations. The paper proposes a MCMC based methods, and then tested the results in some datasets. The paper seems to be on a novel track of goals, but needs further development.

**Summary Of The Review:**

The paper seems still in its preliminary stage, a little bit limited in technical novelties, and more importantly empirical validation. I'm happy to give it a more thorough read if the authors can provide more convincing and more comprehensive empirical evaluations.

---

> ### Author Response · Authors · 2022-11-17
> **Authors Response**
>
> We thank the reviewer for their feedback and address major points below.
>
> * *the algorithm still needs further development, it's hard for me to believe a MCMC based method is the best solution in this scenario, given the rich techniques developed in deep learning regime.*
>
> MCMC methods (e.g. Langevin MC) are used in deep learning in the context of bayesian neural networks and uncertainty quantification among other applications. While Metropolis-Hasting is indeed an old sampling algorithm, it has the advantage that it does not require differentiable transformations. However, nothing precludes our method from being used with other sampling techniques. Finally there are applications where MCMC methods have been found to perform better than deterministic gradient approaches, for example in adversarial robustness.
>
> * *the empirical scope is quite limited: the deepAA method is barely compared.*
>
> We have now included DeepAA in all of the settings in table 1. We had not done it because the authors had not reported results in those settings. We have also included additional experiments that highlight the informativeness of dual variables using synthetic datasets (Appendix C.3).
>
> * *the paper needs further development on its empirical end to be considered for publication.*
>
> We have included additional experiments in section 5 and appendix C.2 and C.3.
>
> * *the MCMC based method can probably be further improved in terms of efficiency and performances with more modern techniques.*
>
> We appreciate the feedback and agree that improving sampling is a relevant research direction. However, we want to point out that MCMC methods are still actively used in many areas, and we give empirical evidence that they work well in the problem we are tackling.
>
> * *Algorithms 1 and 2 seem fairly standard to me, not sure where the novelty is from.*
>
> We want to emphasize that our contribution is to show that data augmentation can be formulated as an invariance constrained learning problem, and that this problem can be solved using standard primal-dual and MCMC methods (algorithms 1 and 2). It is not the algorithms themselves that are novel but, to the best of our knowledge, their connection and application to this problem - and the problem formulation itself - is indeed novel.

---

### Official Review · Reviewer_8185 · 2022-10-24

**Confidence:** 3
**Correctness:** 4
**Technical Novelty And Significance:** 3
**Empirical Novelty And Significance:** 3
**Recommendation:** 6

**Clarity, Quality, Novelty And Reproducibility:**

The paper is clearly written and is of high theoretical quality. The quality of the experiments is a bit lower. The method seems novel and the experiments seem reproducible.

**Strength And Weaknesses:**

Strengths:
- The problem of tuning data augmentation to different tasks is important.
- The paper is clearly written and the theory seems correct.

Weaknesses:
- The experiments are a bit limited and inconclusive.

Major comments:
- How does the choice of just two MC steps in the experiments affect the performance? I would imagine that the samples stay quite close to the uniform proposal then instead of actually sampling from the right distribution.
- DeepAA seems like a quite strong baseline, it outperforms all other methods in one out of two experiments where it is included. Why is it not included in the other ones? I feel like it should be.
- Given that TA is designed to be a ridiculously simple approach, it performs quite well compared to the proposed method. How does it compare in terms of runtime? I would imagine that TA would be much faster, is that true?
- I appreciate the image classification experiment, but I think it would be illustrative to have an experiment with artificial invariances (similar to the ones in [1]), so one could discern to what extent the learned augmentations would correctly capture those.

Minor comments:
- The related work could mention [1]
- In Eq. 3, what is $h$? Should that be $f_\theta$?
- Sec. 3.2: a intance -> an instance

[1] https://arxiv.org/abs/2202.10638

**Summary Of The Paper:**

The authors propose to treat data augmentation in the context of a constrained optimization problem, where the constraints are defined using a set of desired invariances. The data augmentations are then chosen using an MCMC scheme based on the loss of the model (which is supposed to be invariant), while the strength of their influence is optimized using a Langrage multiplier.

**Summary Of The Review:**

Overall, I think this is a really nice idea with a strong theoretical motivation. If the experiments could be improved, I would be happy to recommend acceptance; for now, I'm keeping my score slightly lower due to the inconclusive experiments.

UPDATE: Based on the changes made by the authors during the rebuttal, I have updated my score.

---

> ### Author Response · Authors · 2022-11-17
> **Authors' Response**
>
> We thank the reviewer for their feedback. We have corrected all minor comments, and address the major comments below.
>
> * *How does the choice of just two MC steps in the experiments affect the performance? I would imagine that the samples stay quite close to the uniform proposal then instead of actually sampling from the right distribution.*
>
> Appendix C.3 shows how increasing the number of steps allows the samples to deviate further from uniformity and thus results in lower entropy augmentation distributions and larger dual variables (Figures 5 and 6). Yet, we have found it has little impact in terms of classification accuracy, as now highlighted at the end of section 5 (shown by Figure 3). In summary, the accuracy with which we sample $\lambda^{\star}_c$ does not have a large impact on performance.
>
> * *DeepAA seems like a quite strong baseline, it outperforms all other methods in one out of two experiments where it is included. Why is it not included in the other ones? I feel like it should be.*
>
> We have now included DeepAA in all of the settings in Table 1. We had not done it because the authors had not reported results in those settings and running their method using the reported parameters lead to worst performance than TA. We have now run their method on all settings using the reported hyperparameters and publicly available code.
>
> * *Given that TA is designed to be a ridiculously simple approach, it performs quite well compared to the proposed method. How does it compare in terms of runtime? I would imagine that TA would be much faster, is that true?*
>
> We have added a runtime analysis in appendix C.1.5. The added computational cost of our algorithm is that of computing a forward pass for each MCMC step, so using two sampling steps, our method takes roughly $2 \times$ Trival Augment's training time. These times correspond solely to training, and it is difficult to account for the time taken to tune the hyperparameters of each method, which hinders direct comparisons. In the case of DeepAA (Zheng et al, 2022), it requires running a data augmentation policy search that takes 11 hours (more than $5 \times$ training time) using the same hardware.
>
> * *I appreciate the image classification experiment, but I think it would be illustrative to have an experiment with artificial invariances (similar to the ones in [1]), so one could discern to what extent the learned augmentations would correctly capture those.*
>
> We have added experiments with artificial invariances in section 5.2, similar to the ones in the reference.

---

> > ### Comment · Reviewer_8185 · 2022-11-27
> > **Thanks**
> >
> > Thanks for the changes, I have updated my score.

---

### Official Review · Reviewer_E3e9 · 2022-11-03

**Confidence:** 3
**Correctness:** 3
**Technical Novelty And Significance:** 3
**Empirical Novelty And Significance:** 2
**Recommendation:** 6

**Clarity, Quality, Novelty And Reproducibility:**

The presentation is very clear and didactically well designed. While the proposed approach seems to be novel for learning data augmentation strategies, its benefits could be clarified and also be subject of more extensive experiments. Currently, the practical impact of the approach seems minor. Importantly, also an analysis of the computational complexity of the proposed method is missing (and could present a considerable downside of the approach).

**Details Of Ethics Concerns:**

I have not ethical concerns.

**Strength And Weaknesses:**

Strengths:
+ The overall presentation is clear, the background is explained in detail, and the respective terms and their high level interpretation are discussed.
+ The proposed data augmentation learning method seems to outperform the baselines on CIFAR10 and CIFAR100.
+ The presented problem formulation has the advantage that if a solution to the unconstrained problem is feasible, the presence of that constraint has no effect on the statistical problem.
-> Could this fact also be helpful to assess whether the learned transformations are consistent with the data distribution or impose an unwanted bias?

Weaknesses and open questions:
- Competing papers (e.g., Müller and Hutter, 2021) compare run times of different augmentation algorithms. Such an analysis is completely missing.
- The authors make multiple claims regarding the advantages of their formulation as constraint optimisation problem but these are not demonstrated experimentally.
(Can invariances be learned?)
- In contrast to competing methods, no experiments on ImageNet are provided. This is relevant for the question whether the proposed data augmentation method scales to large, complex datasets.
- The authors claim themselves that they fail to achieve large improvements in accuracy over baselines but this could be attributed to a stagnation in data augmentation research (Müller and Hutter, 2021) and
could reflect limits of the benchmarking setup. In this case, the authors could also look for other datasets and demonstrate improvements of their method over benchmarks there.
- A general question for this line of research: Why are no synthetic experiments conducted to test the ability of automated data augmentation algorithms to recover symmetries in the data? It would be easy to generate data that is invariant to some transformations and test how many samples are required to learn the correct transformations. The claim that the relevant transformations are learned is otherwise unsupported.
- The data augmentation proposal seems to rely on a couple of hyper parameters that need to be tuned in practice.
- A couple of hand-crated baselines are discussed in the introduction but not compared with in the experiment section.

Points of minor critique and open questions:
- No guarantees with respect to the convergence of the sampler to the equilibrium distribution are provided. (This would, however, be a hard problem and might be too much to ask for.)
- The related work section is quite short and adds little value in addition the introduction.
- Figures 1 and 2 are a bit difficult to read for yellow markers.
- Considering that a lot space is left in the main paper, the conclusions and insights into the proposed method could be extended. In particular, moving the analysis of the sampled transformations could highlight additional features of the proposed method.


**Summary Of The Paper:**

The authors propose a novel approach to learn a data augmentation method by formulating the task as invariance-constrained learning problem. They leverage Monte Carlo Markov Chain (MCMC) sampling to solve it, which achieves state-of-the-art results on CIFAR10 and CIFAR100 for specific wide neural network architectures.


**Summary Of The Review:**

While the paper is well motivated and clearly written, some relevant analysis (like the runtime analysis, potentially additional experiments that highlight the specific advantages of the proposed approach, etc.) are missing.

---

> ### Author Response · Authors · 2022-11-17
> **Authors' Response**
>
> We want to thank the reviewer for their efforts and feedback. Find our reply to the points raised below.
>
>
> * *Competing papers (e.g., Müller and Hutter, 2021) compare run times of different augmentation algorithms. Such an analysis is completely missing.*
>
> We have added a runtime analysis in appendix C.1.5. These times correspond solely to training, and it is difficult to account for the time taken to tune the hyperparameters of each method, which hinders direct comparisons. In the case of DeepAA (Zheng et al., 2022), it requires running a data augmentation policy search that takes 11 hours (more than $5 \times$ training time) using the same hardware.
> Note that all of our experiments use only two steps of the MH sampler.
>
> * *The authors make multiple claims regarding the advantages of their formulation as constraint optimisation problem but these are not demonstrated experimentally. (Can invariances be learned?)*
>
> The goal of our paper is not to learn invariances present in the distribution but to impose it. That being  said,  dual variables do give information about how hard it is to satisfy an invariance in terms of the loss on the original data. We have added experiments that demonstrate this (now on Appendix C.3) on datasets with synthetic invariances.
>
> * *In contrast to competing methods, no experiments on ImageNet are provided. This is relevant for the question whether the proposed data augmentation method scales to large, complex datasets.*
>
> We agree, that is indeed an open question. However, these experiments exceed our current computational budget. We, therefore, leave them for future work.
>
> * *The authors claim themselves that they fail to achieve large improvements in accuracy over baselines but this could be attributed to a stagnation in data augmentation research (Müller and Hutter, 2021) and could reflect limits of the benchmarking setup. In this case, the authors could also look for other datasets and demonstrate improvements of their method over benchmarks there.*
>
> We agree with the reviewer. Unfortunately, other than Imagenet, there are no other widespread benchmark datasets in automatic data augmentation literature, to the best of our knowledge. Designing a new experimental set-up, and more importantly tuning not only our method but also baselines to allow for a fair and meaningful comparison is undoubtedly a valuable and timely research endeavor, but beyond the intended scope of this paper.
>
> * *A general question for this line of research: Why are no synthetic experiments conducted to test the ability of automated data augmentation algorithms to recover symmetries in the data? It would be easy to generate data that is invariant to some transformations and test how many samples are required to learn the correct transformations. The claim that the relevant transformations are learned is otherwise unsupported.*
>
> As already mentioned, we have added experiments on datasets with synthetic invariances in the Appendix C3. We want to emphasize that the goal of our paper is not to learn invariances present in the distribution but to impose them on predictors (i.e. solve CSRM). We do highlight that dual variables give information about how hard it is to satisfy an invariance in terms of the loss on the original data, and that this can reflect invariances present in a learning task. In that sense, we demonstrate empirically that dual variables that correspond to synthetic invariances in a dataset are considerably lower than those associated with other transformations (See table 10 in Appendix C.3).
>
> * *The data augmentation proposal seems to rely on a couple of hyper parameters that need to be tuned in practice.*
>
> We show in the experimental section that our approach is not overly sensitive to the constraint level epsilon (figure1). We also provide ablations on the number of the MCMC sampler steps at the end of section 5 (figure 3). In both cases, we show that model performance is not overly sensitive to these hyperparameters.
>
> * *A couple of hand-crated baselines are discussed in the introduction but not compared with in the experiment section.*
>
> We consider that (Müller and Hutter, 2021) is a handcrafted baseline, but we are not sure what other baselines the reviewer is referring to. Nevertheless, we welcome new setups to experiment with and compare our approach.

---

> > ### Author Response · Authors · 2022-11-17
> > **Response to Points of minor critique and open questions**
> >
> > * *The related work section is quite short and adds little value in addition the introduction.*
> >
> > We have moved it to the appendix and moved more experimental analyses from the appendix to the main body.
> >
> > * *Figures 1 and 2 are a bit difficult to read for yellow markers.*
> >
> > We have now changed the colors in both figures.
> >
> > * *Considering that a lot space is left in the main paper, the conclusions and insights into the proposed method could be extended. In particular, moving the analysis of the sampled transformations could highlight additional features of the proposed method.*
> >
> > We have moved some of the analysis of the sampling step ablation to the main body, and have now approached the page limit.

---

> > ### Comment · Reviewer_E3e9 · 2022-12-06
> > **Acknowledgement of changes and response**
> >
> > I thank the authors for their detailed response and change of the manuscript.
> >
> > I appreciate the additional experiments in the appendix on synthetically transformed data.
> > While the authors have emphasized that the goal of their paper is not to learn invariances present in the distribution but to impose them, they still want to learn the optimal data augmentation strategy. The question remains therefore what an optimal strategy would be. I would argue that it should be able to generalize invariances in the underlying data distribution based ideally on a small amount of samples. (The dependence on the sample size was not checked by the way.)
> >
> > However, I still believe that the manuscript is maximally marginally above the acceptance threshold for the following reasons:
> >
> > a) The experimental evidence is relatively weak in comparison to what seems to be common in the field. Data augmentations are usually more effective on complex (often large scale) datasets. As many researchers face computational limitations, I actually do not expect results on ImageNet (which are not presented). However, significant performance improvements on some relevant datasets are necessary. If this requires major additional tuning work on new datasets then this hints towards a great disadvantage of the general approach.
> >
> > b) The run time of the proposed approach is considerably worse in comparison with competing methods. Based on the presented experimental evidence (and without any given theoretical run time statement) it is unclear if the method would even scale towards larger datasets.

---

> > > ### Author Response · Authors · 2022-12-07
> > > **Response to Reviewer E3e9**
> > >
> > > We thank the reviewer for their feedback, an address their concerns bellow.
> > >
> > > a) We believe that data augmentation is valuable even in smaller scale datasets. Evidence of that is the fact that CIFAR is still a common benchmark in the field  (Ho et al., 2019; Lim et al., 2019; Hataya et al., 2020; Zhang et al., 2020; Cubuk et al., 2020; LingChen et al., 2020; Zhang et al., 2020; Müller and Hutter, 2021; Zhou et al., 2021b; Zheng et al., 2022; Cheung and Yeung, 2022). In particular, using these smaller scale datasets allows for more complex ablation studies that in our case illustrate several advantages of our method beyond test accuracy. Nevertheless, we commented on ImageNet only because the reviewer mentioned it. We are happy to run additional experiments if they have a suggestion for a larger dimensional, smaller scale (similar to CIFAR) dataset.
> > >
> > > That being said, our goal is not necessarily to improve state-of-the-art test performance, but to learn invariant predictors. Still, our method displays an accuracy improvement of 0.8% over other techniques in CIFAR10 using WideResnet-40-2, which is significant considering the already high performance of regular training (96%) that leads to small performance improvements (i.e. around 0.2% for other methods).
> > >
> > > b) The increased training time of our method comes not only with an improvement in performance, but also with the additional sensitivity information contained in the dual variables and, most importantly, (approximate) invariance of the learned predictor (given the constrained learning results detailed in 4.1). No other data augmentation method claims such a guarantee.
> > >
> > > Additionally, it is worth noting that our method requires very little tuning as opposed to SOTA approaches such as DeepAA, whose initial search phase is almost 3x slower than our training. Finally, it is worth noting that these relations will typically hold in higher dimensions, as our method typically involves a few additional forward passes through the network.

---

### Official Review · Reviewer_wrTV · 2022-11-03

**Confidence:** 5
**Correctness:** 1
**Technical Novelty And Significance:** 2
**Empirical Novelty And Significance:** 1
**Recommendation:** 5

**Clarity, Quality, Novelty And Reproducibility:**

Writing can be improved with a more streamlined and concise presentation. Some aspects of the framework (e.g., semi-infinite constrained learning and MCMC sampler required) exist in a previous work in adversarial robustness.

**Strength And Weaknesses:**

Strengths:

1. Automatically learning data augmentations for a particular task is a significant and important challenge which helps not just in-distribution but also for out-of-distribution robustness of models.
2. Not requiring differentiable transformations (unlike baselines) can be helpful.
3. Typically, the tradeoff between the original task loss and the augmentation loss is done via hyperparameter tuning, the proposed approach instead maximizes the relevant dual variable (whose advantages are discussed in the paper).

Weaknesses:

1. I think the paper can be improved with a more formal treatment of the problem and method.
   1. The problem is not formally defined, for example, the paper never discusses formally the true invariances of the data distribution. Essentially, the task of an automatic data augmentation method will be to learn transformations that hold for the true (unknown) data generation process.
   2. Degree of invariance and approximate invariance are used loosely and not well-defined.
   3. The paper is motivated solely on improving the sample complexity but no such bounds are provided; while data augmentation itself can lower the sample complexity, it is unclear if that continues to hold when learning augmentations automatically especially with an approximation from constrained optimization (CSRM) to unconstrained optimization (D-CERM).

2. Empirical evaluation does not show case the claims of advantages of the proposed method vs the baselines.

   1. Experiments do not explicitly show cases when data augmentation results in biasing the model and is harmful, which is one of the main reasons for learning the appropriate distributions over the transformations.

   2. Another advantage of the proposed method compared to the baselines is to allow non-differentiable transformations, but no such experiments are performed.
   3. Overall, the empirical gain in performance is not significant compared to the baselines.



Other Questions/Comments:

1.  Introduction: “The use of invariance overcomes the need to search for a specific distribution over transformations”. I do not understand this statement.
2. As opposed to the existing methods, $\gamma$ is treated as a dual variable (and optimized) which allows the tradeoff to be learnt directly. However, $\epsilon$ is a hyperparameter which is cross-validated which also trades-off the approximate invariance, thus leading to the same problem. Further, it is not clear from the paper, what cross-validation methodology is used (i.e., what is the training and test datasets for this cross-validation)?
3. There seems to be a single slack variable \epsilon that controls all different types of transformations (for example, rotations, translations, shear, etc) with no additional regularization to force toward invariance unlike [1].

3. I do not agree with the statement “sampling $\lambda^*_c$ accurately is not a concern given that we are interested in promoting approximate invariance” as we still need to characterize the kind of approximation. For instance, using a constraint level ($\epsilon$) is one way of approximation but it is different from not able to accurately estimating $\lambda_c^*$.

Minor/Typos:

1. Typos: “whithout”, “infromativeness”

2. $\lambda^*_c$ takes different inputs at different places in Section 4.2.


**References**
[1] Benton, Gregory, et al. "Learning invariances in neural networks from training data." Advances in neural information processing systems 33 (2020): 17605-17616.



**Summary Of The Paper:**

Paper proposes to automatically learn which transformations to augment during training without introducing biases, for example, when the augmentations are not appropriate for the task. Authors rewrite the problem as a semi-infinite constrained problem similar to a recent adversarial robustness framework. Proposed approach learns to sample from a distribution over the transformations using a Metropolis-Hastings approach.

**Summary Of The Review:**

Paper can be improved with a more formal treatment of the problem and method (e.g., in defining the problem, discussing sample complexity of the proposed method). Empirical evaluation does not show case the claims of advantages of the proposed method vs the baselines (tasks where some augmentations are not appropriate, non-differentiable transformations).

---

> ### Author Response · Authors · 2022-11-17
> **Authors Response**
>
> We thank reviewer wrTV for the detailed and thoughtful feedback and suggestions. We hope that the clarifications, changes and additional experiments described below address the reviewers concerns.
>
> *1.1.  The problem is not formally defined, for example, the paper never discusses formally the true invariances of the data distribution. Essentially, the task of an automatic data augmentation method will be to learn transformations that hold for the true (unknown) data generation process.*
>
> The problem we tackle is formally defined in section 3.2 (CSRM), and is motivated on section 3.1. Section 4 then shows how to solve this problem empirically, which is the main focus of the paper. We agree with the reviewer, that the true invariances of the data distribution are not formally addressed. To clarify this, we have added a paragraph at the end of section 2 explaining that, unlike invariance learning, we do not seek to learn transformations from the data, but to impose invariance on the predictor.
>
> *1.2. Degree of invariance and approximate invariance are used loosely and not well-defined.*
>
> The notion of invariance we refer to throughout the text is explicitly defined in Section 3.1, in equation 5, in terms of the expected loss. We acknowledge that the notion of invariance of the predictor we address is not standard and thus prone to confusion. We have added a sentence to clarify, and a section (Appendix D) that discusses different notions of invariance and their relation to our work.
>
> *1.3 The paper is motivated solely on improving the sample complexity but no such bounds are provided; while data augmentation itself can lower the sample complexity, it is unclear if that continues to hold when learning augmentations automatically especially with an approximation from constrained optimization (CSRM) to unconstrained optimization (D-CERM).*
>
> We agree with the reviewer’s paper summary, improving sample complexity is not the only rationale for promoting invariance (e.g., robustness or out of distribution performance). We have clarified this at the end of section 2.
> Indeed, our work does not focus on analyzing the benefits of invariance, but on how to impose it empirically/operationally, i.e. how to solve CSRM. Our proposed solution relies on sample complexity results from constrained learning theory (Chamon et al., 2021) that show that the approximation of the constrained problem (D-ECRM) has the same sample complexity as unconstrained learning.
>
> *2.1. Experiments do not explicitly show cases when data augmentation results in biasing the model and is harmful, which is one of the main reasons for learning the appropriate distributions over the transformations.*
>
> We have now added an experiment to address this shortcoming in Appendix C.2. In summary, we apply data augmentation by sampling transformations according to $\lambda^{\star}_c$ and find that it decreases test accuracy from 78% to 75%, when compared to training  without augmentation. Moreover, we show that training using both $\lambda^{\star}_c$ as an augmentation distribution and the original data (i.e., adding the identity) with equal weights, improves accuracy to 80%.
>
>
> *2.2 Another advantage of the proposed method compared to the baselines is to allow non-differentiable transformations, but no such experiments are performed.*
>
> In the experiments in Section 5, four out of the sixteen transformations used are non-differentiable, we have now made this explicit (Section 5 paragraph 1).
>
> *2.3 Overall, the empirical gain in performance is not significant compared to the baselines.*
>
> (i) It is true that our approach has little empirical gains however, our approach has other advantages rather than empirical performance as explained at the end of section 4, such as  informativeness/interpretability of dual variables, and the use of non-differentiable transformations. (ii) Recently there have been no substantial improvements in this benchmark setting, which indeed  is an issue that automatic data augmentation research is facing. Although not by a large margin, to the best of our knowledge this is the first work that outperforms Trivial Augment wide (Müller and Hutter, 2021) in CIFAR datasets. We have now updated Table 1 to include a more thorough comparison with baseline methods in all settings.

---

> > ### Author Response · Authors · 2022-11-17
> > **Response to other Questions/Comments**
> >
> > *1. Introduction: “The use of invariance overcomes the need to search for a specific distribution over transformations”. I do not understand this statement.*
> >
> > This statement is referring to section 3.1, where we define a notion of invariance and show that it leads to an augmentation distribution. By choosing the transformation set $\mathcal{G}$ and the level of invariance, which is controlled by $\epsilon$, our approach adapts the augmentation distribution to the data and model, without the need to manually choose/search or parametrize a distribution over transformations.
> > We have now extended this statement in the text. Hopefully, it is now clearer, if the reviewer considers it remains unclear we can expand.
> >
> > *2. As opposed to the existing methods, lambda is treated as a dual variable (and optimized) which allows the tradeoff to be learned directly. However, epsilon is a hyperparameter which is cross-validated which also trades-off the approximate invariance, thus leading to the same problem. Further, it is not clear from the paper, what cross-validation methodology is used (i.e., what is the training and test datasets for this cross-validation)?*
> >
> > $\epsilon$ is indeed a hyperparameter that needs to be adjusted. However, its value depends on the task rather than on the data. Since $\epsilon$ should depend on the value of the statistical problem CSRM, we use cross validation to select it.
> > As noted in the paper, the main advantage is that it is interpretable and, unlike the penalization required to achieve  a desired level of invariance, $\epsilon$ does not depend on the number of samples.
> > We want to highlight that our work focuses on how to solve CSRM for a fixed value of $\epsilon$. Also, the ablation experiment on the constraint level (see Figure 1 on section 5) shows that our approach is not overly sensitive to this hyperparameter.
> > Lastly, in our experiments we selected epsilon using 10% of the training data (randomly sampled) as a validation set, and searched for epsilon over a grid, as detailed on appendix B.3.
> >
> > *3. There seems to be a single slack variable \epsilon that controls all different types of transformations (for example, rotations, translations, shear, etc) with no additional regularization to force toward invariance unlike [1].*
> >
> > We restricted ourselves to only one slack variable throughout the paper, for clarity of presentation and because it worked well empirically. With minimal modifications, our approach can be extended to accommodate any number of constraints specified by transformation sets $\mathcal{G}_i$ and constraint levels $\epsilon_i$. We have added this comment in section 3. As an example, additional experiments on appendix C.3 (with synthetic invariances) now portray such cases.
> > Regarding the second point, our approach allows imposes invariance without the need of a regularization term by deriving an explicit augmentation distribution ($\lambda^{\star}$) and showing how it is linked to invariance and the CSRM problem.
> >
> > *4. I do not agree with the statement “sampling $\lambda^{\star}_c$ accurately is not a concern given that we are interested in promoting approximate invariance” as we still need to characterize the kind of approximation. For instance, using a constraint level (ϵ) is one way of approximation but it is different from not able to accurately estimating  $\lambda^{\star}_c$.*
> >
> > Indeed, there could be an impact of the sampling approximation accuracy on the performance of the solution. In order to investigate the empirical implications of this approximation on the obtained solutions we provide ablations on section 5 and appendix C.1. More specifically, Figure 1 shows the impact of changing epsilon for a fixed number of sampling steps, while figure 3 shows the effect of changing the number of steps in the MCMC sampler which does not affect test accuracy significantly. Additionally, it is worth pointing out that ultimately our goal is not to sample $\lambda^{\star}_c$ exactly but to solve CSRM. However, the reviewer has a good point, and to avoid confusions about the types of approximations we are referring to we have removed the sentence.

---

> > > ### Author Response · Authors · 2022-11-17
> > > **Connection to previous work in adversarial robustness**
> > >
> > > *Writing can be improved with a more streamlined and concise presentation. Some aspects of the framework (e.g., semi-infinite constrained learning and MCMC sampler required) exist in a previous work in adversarial robustness.*
> > >
> > > The reviewer has a point, but making the connection between adversarial robustness and invariance induced data augmentation explicit is, in our view, one of the contributions of this paper. It is worth noting that these two applications have different goals and properties.
> > > For example, in the context of adversarial robustness invariance is not a property of the data distribution but a desideratum, which results in a trade-off with test performance. Therefore, less attention is paid to generalization and the informativeness of dual variables, as we do in the current work. Furthermore, unlike the aforementioned work, we minimize the statistical risk under a constraint the on risk of transformed samples, and not contrariwise. As explained in the paper, this has the added benefit that if a solution is strictly feasible, it is still optimal for the statistical risk minimization problem.

---

> > > ### Comment · Reviewer_wrTV · 2022-12-03
> > > **Response to authors**
> > >
> > > Thank you for providing clarifications and including additional experiments. I understand better the interpretability aspect of the dual variables from the rebuttal and synthetic experiments in C.3. I have updated my score, but have a few remaining concerns.
> > >
> > > **1.1-1.3:** Since the notion of invariance is different, I believe its benefits should be studied/discussed. Sample complexity argument for data augmentation uses the fact that the true data distribution is invariant to the group of transformations. However, I am not aware of a result that discusses sample complexity while learning a distribution over the given set of transformations. Thus, it is not immediately clear to me how this improves sample complexity. A more formal discussion of the properties of true data distribution assumed would help.
> > > On the other hand, if robustness or out-of-distribution performance are the goals (as mentioned in the response), I think it should be focused more in the paper with experiments showcasing this ability.
> > >
> > > I think the statement "Unlike invariance learning (Jebara, 2003; Zhou et al., 2021a; Benton et al., 2020; Immer et al., 2022), we do not seek to learn the transformations G from the data." is not accurate: Benton et al., (2020) learn a distribution over a given set of transformations as well.
> > > Thanks for clarifying that sample complexity of D-CERM and unconstrained learning is the same, it would be helpful to add in the paper.
> > >
> > >
> > > **2.1:** Maybe I have misunderstood the experiment in Appendix C.2, but it seems to show the opposite of what we would like from the proposed approach.
> > >
> > > My suggested hypothesis to be tested was as follows: Randomly sampling data augmentation with widest range for transformations performs bad in a task, but the distribution learnt by $\lambda^*_c$ performs better. This would substantiate the claim  “…indiscriminate use of data augmentation can introduce biases that outweigh its benefits. This work tackles these issues by automatically adapting the data augmentation while solving the learning task”.
> > >
> > > However, experiment in C.2 suggests that $\lambda^*_c$ is unable to find a good distribution over data augmentation (performing worse even than no augmentation).

---

> > > > ### Author Response · Authors · 2022-12-06
> > > > **Response to Reviewer wrTV**
> > > >
> > > > We thank the reviewer for engaging in discussion and providing insightful comments. We address them bellow.
> > > >
> > > > *1.1-1.3: Since the notion of invariance is different, I believe its benefits should be studied/discussed. Sample complexity argument for data augmentation uses the fact that the true data distribution is invariant to the group of transformations. However, I am not aware of a result that discusses sample complexity while learning a distribution over the given set of transformations. Thus, it is not immediately clear to me how this improves sample complexity. A more formal discussion of the properties of true data distribution assumed would help. On the other hand, if robustness or out-of-distribution performance are the goals (as mentioned in the response), I think it should be focused more in the paper with experiments showcasing this ability.*
> > > >
> > > > We want to begin by clarifying that we do not learn a distribution over transformations. We find the distribution that induces the predictor to be invariant on the sampling set. That distribution is deterministic, and its relation to the distribution that would impose invariance on the population is certainly interesting, but is beyond the scope of this work. There are already results relating imposing invariance over the sampling set to reductions in sample complexity. That being said, studying potential sample complexity gain with respect to our notion of invariance would also be interesting. However, we want to emphasize that providing sample complexity results under distributional invariance assumptions is out of the scope of this paper, which focuses on how to impose invariance on the predictor operationally.
> > > >
> > > > *I think the statement "Unlike invariance learning (Jebara, 2003; Zhou et al., 2021a; Benton et al., 2020; Immer et al., 2022), we do not seek to learn the transformations G from the data." is not accurate: Benton et al., (2020) learn a distribution over a given set of transformations as well.*
> > > >
> > > > This is a good point. Indeed, some of the invariance learning literature uncovers invariances present in the data by learning an augmentation distribution. Benton et. al (2020) learns the support of a uniform augmentation distribution from the data. However, their main aim is to recover a subset of the initial search space corresponding to “true” invariances of the data distribution. On the other hand, our main goal is to impose invariance over a given set of transformations on the predictor. For example, once a set of invariant transformations is learned, our approach can be used to impose this invariance on predictors (as opposed to using a uniform distribution over those same transformations). If the reviewer feels it is clearer, we would be happy to change that sentence to "Unlike invariance learning (Jebara, 2003; Zhou et al., 2021a; Benton et al., 2020; Immer et al., 2022), we do not seek to learn from samples the set G of invariances of the data. Instead, our goal is to learn a predictor that is (approximately) invariant to a given set of transformations G.
> > > >
> > > > *Thanks for clarifying that sample complexity of D-CERM and unconstrained learning is the same, it would be helpful to add in the paper.*
> > > >
> > > > Thanks for the suggestion, we will add it in future versions of the paper.

---

> > > > > ### Author Response · Authors · 2022-12-06
> > > > > **Response to Reviewer wrTV (continues)**
> > > > >
> > > > > *2.1: Maybe I have misunderstood the experiment in Appendix C.2, but it seems to show the opposite of what we would like from the proposed approach.
> > > > > My suggested hypothesis to be tested was as follows: Randomly sampling data augmentation with widest range for transformations performs bad in a task, but the distribution learnt by Λc∗ performs better. This would substantiate the claim “…indiscriminate use of data augmentation can introduce biases that outweigh its benefits. This work tackles these issues by automatically adapting the data augmentation while solving the learning task”. However, experiment in C.2 suggests that λc∗ is unable to find a good distribution over data augmentation (performing worse even than no augmentation).*
> > > > >
> > > > > In this case, we misunderstood your point. From your comments it would seem that your point was that sampling uniformly over a poorly defined set of transformations could hinder performance, while our point was that even if the set of transformations is chosen appropriately for the task, this does not mean that *any* augmentation distribution is beneficial. The fact is that both the set $\mathcal{G}$ and the distribution $\mathfrak{G}$ need to be chosen appropriately. We focus on the latter since we do not seek to learn the invariant transformations: we assume they are given.
> > > > >
> > > > > We want to clarify that in our approach, $\lambda^{\star}_c$ *is not* the distribution of training samples. The training distribution is a mixture of data augmented according to $\lambda^{\star}_c$ and the original data (or equivalently, sampling the identity transformation $e$), weighted by the dual variable $\gamma$. That is, explicitly our data is sampled from $(\frac{1}{1+\gamma})e + \frac{\gamma}{1+\gamma}  \lambda^{\star}_c$.  Our experiments in section 5 do confirm that this training distribution, in which $\lambda^{\star}_c$ and $\gamma$ are adjusted while solving the learning task, can outperform uniform sampling.
> > > > >
> > > > > The claim addressed by the experiments in appendix C.2 is that certain distributions over transformations can introduce biases and hinder performance. We illustrate this by training only with data augmented by sampling transformations from $\lambda^{\star}_c$), showing that it performs worse than training on the original data (without augmentation). The point is that by adjusting the amount of augmentation (by including the original data - or equivalently putting mass at the identity transformation) we can overcome these biases. We will remove the term “indiscriminate”.
> > > > >
> > > > > Thanks again for your time and efforts.

---

### Author Response · Authors · 2022-11-17
**Thanks to reviewers and initial response**

We sincerely thank all reviewers for their efforts in reviewing our paper, and their valuable insights and feedback. We have carefully addressed your comments and suggestions, and hope that the forthcoming exchanges and discussion can lead to further improvements  in the paper.

We provide a summary of the main changes we have made to our paper during this discussion period based on the reviewers' input:

* Added clarification that our approach does not aim to learn invariances in the data but impose them, in sections 1 and 2. (Reviewers wrTV, E3e9 and 8185)
* Added comparison with DeepAA in all setups  (Reviewers 8185 and 159N)
* Added Runtime analysis on appendix C.2 (Reviewers E3e9 and 8185)
* Added experiments with synthetic invariant datasets on appendix C.3 (Reviewers wrTV, E3e9 and 8185)
* Added an example where data augmentation is harmful in appendix C.2 (Reviewer wrTV)
* Moved MC sampler steps ablation to main body (Reviewers wrTV and E3e9).
* Fixed typos and other minor comments.

---

### Decision · Program_Chairs · 2023-01-20

**Decision:**

Reject

**Justification For Why Not Higher Score:**

The support for acceptance is not strong enough.


**Justification For Why Not Lower Score:**

N/A


**Metareview: Summary, Strengths And Weaknesses:**

We appreciate the authors for responding to the review comments, adding more experiments, and engaging in discussion with some reviewers. Automating the otherwise manual data augmentation process is very useful and this work has contributed interesting ideas to this topic. Most of the existing automated data augmentation methods are so computationally demanding that it is simply infeasible to adopt them for practical applications. Gradient-based methods are computationally more attractive than others, such as those based on reinforcement learning, although they still cannot be considered as lightweight methods. The fact that the proposed method is based on MCMC sampling alerts us that efficiency, and hence practicality, is an important aspect that is worth considering. In their response to one of the reviewers, the authors admit that experiments involving larger datasets such as ImageNet exceed their current computational budget. This is in line with our concern that the proposed method may not be computationally feasible for practical applications. Among other comments and concerns, the authors are recommended to pay more attention to this aspect in further enhancing their method that hopefully can really be adopted by practitioners to make real-world impact.